# Active Learning via Classifier Impact and Greedy Selection for Interactive Image Retrieval

**Leah Bar**                                                                 *barleah.libra@gmail.com*
*Department of Applied Mathematics*
*Tel-Aviv University, Israel*

**Boaz Lerner**                                                              *boazl@originai.co*
*OriginAI, Ramat-Gan, Israel*

**Nir Darshan**                                                              *nir@originai.co*
*OriginAI, Ramat-Gan, Israel*

**Rami Ben-Ari**                                                             *ramib@originai.co*
*OriginAI, Ramat-Gan, Israel*

**Reviewed on OpenReview:** *https://openreview.net/forum?id=b68QOenPWy*

## Abstract

Active Learning (AL) is a user-interactive approach aimed at reducing annotation costs by selecting the most crucial examples to label. Although AL has been extensively studied for image classification tasks, the specific scenario of interactive image retrieval has received relatively little attention. This scenario presents unique characteristics, including an open-set and class-imbalanced binary classification, starting with very few labeled samples. We introduce a novel batch-mode Active Learning framework named GAL (Greedy Active Learning) that better copes with this application. It incorporates new acquisition functions for sample selection that measure the impact of each unlabeled sample on the classifier. We further embed this strategy in a greedy selection approach, better exploiting the samples within each batch. We evaluate our framework with both linear (SVM) and non-linear MLP/Gaussian Process classifiers. For the Gaussian Process case, we show a theoretical guarantee on the greedy approximation. Finally, we assess our performance for the interactive content-based image retrieval task on several benchmarks and demonstrate its superiority over existing approaches and common baselines. Code is available at https://github.com/barleah/GreedyAL.

## 1 Introduction

Annotated datasets are in high demand for many machine learning applications today. Active Learning (AL) aims to select the most valuable samples for annotation, which, when labeled and integrated into the training process, can significantly enhance performance in the target task (e.g., a classifier). In recent years, task-specific AL has gained popularity across various domains, including multi-class image classification Parvaneh et al. (2022); Emam et al. (2021), few-shot learning Ayyad et al. (2021); Pezeshkpour et al. (2020), pose estimation Gong et al. (2022), person re-identification Liu et al. (2019), object detection Wu et al. (2022), and interactive Content-Based Image Retrieval (CBIR) Barz & Denzler (2021); Gosselin & Cord (2008); Mehra et al. (2018); Ngo et al. (2016).

Image retrieval is a long-standing challenge in computer vision, crucial for data mining within large image collections and with applications in fields such as e-commerce, social media, and digital asset management. An interactive image retrieval system learns which images in the database belong to a user's query concept, by analyzing the example images and feedback provided by the user. The challenge is to retrieve the

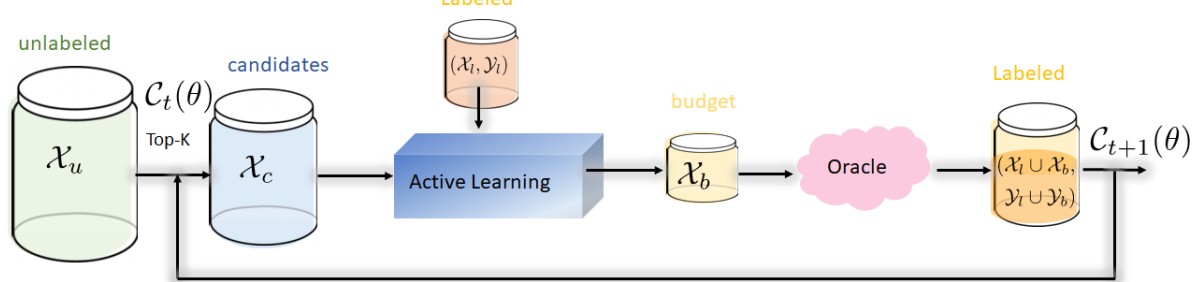

Figure 1: Main flow of the AL cycle. The top-K candidate set at cycle $t$ determined by the classifier $\mathcal{C}_t(\theta)$, can be selected as the pool from the unlabeled/search corpus. The AL module extracts a batch set $\mathcal{X}_b$ which is sent for annotation by a user (oracle) that generates the label set $\mathcal{Y}_b$. Based on the extended training set, a new classifier $\mathcal{C}_{t+1}(\theta)$ is trained for the next cycle.

relevant images with minimal user interaction Qazanfari et al. (2017); Putzu et al. (2020); Banerjee et al. (2018); Roli et al. (2004); Lerner et al. (2023). This process involves iterative feedback, where users provide relevance feedback on a set of images suggested by the model, which is then used to retrain the retrieval model, refining search results over time. Content-Based Image Retrieval (CBIR) relies on visual content for retrieval. In Interactive Image Retrieval (IIR) within CBIR, the search process usually begins with a small set of user-provided query images (typically 1-5), which results in limited data for training an effective retrieval model. Active learning techniques can then be applied through feedback rounds, to identify the most informative images, prompting the user to label them. The system then trains a new retrieval model (typically a classifier), guiding the search results toward the query concept. A general pipeline illustrating the AL process for IIR is shown in Fig. 1.

**Active learning for image classification** has traditionally been the focus of AL methods Karzand & Nowak (2020); Wang et al. (2022); Emam et al. (2021); Xie et al. (2021); Parvaneh et al. (2022); Kothawade et al. (2021); Citovsky et al. (2021), typically focusing on class-balanced, closed-set datasets and often starting with a relatively large number of labeled samples. More recent works address the challenge of starting the AL process with few labeled samples, known as *cold start*. This issue is inherent in IIR use case, introducing a challenge for selecting informative samples and train an effective retrieval model. Some recent approaches address the cold start problem within the context of image classification Hacohen et al. (2022); Yehuda et al. (2022). Others focus on issues like imbalance data Aggarwal et al. (2020); Kothawade et al. (2021) or redundant examples Citovsky et al. (2021). However, these methods are generally not designed or tested for scenarios that combine multiple complexities, as seen in IIR, such as cold start, class imbalance, rare classes, and open-set setting.

**Pool-based Active Learning** is a setting where the learner has access to a large pool of unlabeled data and can iteratively select samples from this pool for labeling. This is particularly useful when labeling is expensive, as it allows the model to focus on samples that are expected to provide the most information for improving performance. IIR naturally follows this strategy and operates within the framework of pool-based AL. In the context of pool-based IIR, early studies have used tuned SVMs with either engineered or deep features Gosselin & Cord (2008); Ngo et al. (2016); Rao et al. (2018); Tong & Chang (2001), leveraging the SVM's strong regularization for small training sets. For instance, Tong & Chang (2001) employed a kernel SVM for binary classification.

**AL in (content-based) interactive image retrieval** has been used to improve retrieval performance, allowing users to find target images with minimal interactions. The system selects critical samples from a pool of unlabeled images, prompting the user to label them as relevant or irrelevant. These labels are then used to train a more accurate retrieval model, iteratively refining its understanding of user intent. In CBIR, the AL process typically follows a pool-based approach Manjunath et al. (2000), where the learner accesses a pool of unlabeled data and requests labels for specific instances. This leads to a binary classification task characterized by imbalanced classes and an open-set scenario (where categories in the search domain are

often unknown). The negative class includes irrelevant images from diverse classes, making the classification asymmetric. Batch Mode AL (BMAL) Karzand & Nowak (2020); Tong & Koller (2001) selects a batch of images at each iteration for user feedback, but traditional AL methods for standard image classification can struggle in this setting due to model instability and unreliable uncertainty estimation Yuan et al. (2020); Jin et al. (2022); Ning et al. (2022).

**Active Learning strategies** aim to evaluate and select unlabeled samples to improve an objective metric, such as classification accuracy or, in our case, retrieval performance. Common criteria for selection include uncertainty Brinker (2003); Sener & Savarese (2018). Uncertainty-based selection focuses on samples near the decision boundary to refine it, but it often overlooks the broader data distribution. Moreover, accurately measuring uncertainty requires a sufficient number of labeled samples, which is often lacking, especially in the early cycles of Interactive Image Retrieval (IIR). In this context, many methods that begin with a cold start—having very few labeled samples—tend to be inefficient and may underperform compared to basic random selection processes Hacohen et al. (2022); Yehuda et al. (2022); Chandra et al. (2021). On the other hand, diversity-based selection aims to represent the entire data distribution, but it can introduce redundant selections or include less informative samples that are far from the decision boundary, offering minimal improvement to the classifier. Recent studies have shown that a hybrid approach, which integrates both uncertainty and diversity, can leverage the advantages of each strategy to yield better results Agarwal et al. (2020); Wang et al. (2016); Yang et al. (2015).

**In this work**, we introduce a novel hybrid AL algorithm specifically designed for Interactive Image Retrieval (IIR). IIR can be framed as a binary classification task with several unique characteristics: (i) Open-set: The number of classes and their categories in the pool are unknown. (ii) Imbalance: Often, less than 1% of the pool contains the query concept (positive class). (iii) Asymmetric sets: The positive set contains a single semantic class, while the negative set can include diverse samples from various categories. (iv) Cold start: Only a few labeled samples are available at each cycle, particularly in the initial and critical cycles. To address these challenges, we propose a Batch Mode Active Learning (BMAL) method for IIR that effectively handles the cold start in an open-set scenario. Our approach introduces acquisition functions that evaluate the global impact of potential samples on the decision boundary for both linear and non-linear classifiers. For SVM and MLP classifiers we evaluate the influence of each possible label (positive or negative), while for Gaussian Processes, we aim to minimize the overall uncertainty of the classifier during sample selection. Additionally, to cope with the scarcity of labeled samples, we introduce a greedy scheme that optimally exploits each sample in the subsequent selection of each batch. This method effectively combines both uncertainty and diversity, as demonstrated in Section 4, providing a robust solution to challenges in IIR. To summarize, we present an innovative approach to Batch Mode Active Learning (BMAL) for IIR tasks with the following contributions:

1. We propose new acquisition functions that quantify the impact value on the classifier as a selection strategy, tailored to both linear and non-linear classifiers. Our framework is adaptable to different classifiers, where, for instance, the impact value can measure the global shift in the decision boundary or the level of global uncertainty of the classifier.

2. We propose a novel greedy scheme to cope with very few labeled samples, focusing on only one class and operating in an open-set regime with highly imbalanced classes.

3. For the Gaussian Process-based classifier, we establish a lower bound on the performance of the greedy algorithm using the $(1 - 1/e)$-Approximation Theorem.

4. We present a more realistic multi-label benchmark for the Content-Based Image Retrieval (CBIR) task, named FSOD, where the query concept involves an object within the input image.

5. We evaluate our framework using three classification methods (linear and non-linear) on four diverse datasets, showcasing superior results compared to previous methods and strong baselines.

## 2 Related Work

Two main characteristics drive the design of AL methods, namely *diversity* and *uncertainty*.

Few works address the batch (budget) size of the selected samples at each cycle of the AL procedure and the cold-start scenario. Recent studies such as Hacohen et al. (2022); Yehuda et al. (2022) have investigated the influence of budget size on active learning strategies and have also addressed the challenge of cold start, where the initial labeled training set is small Hacohen et al. (2022); Yehuda et al. (2022); Yuan et al. (2020); Gao et al. (2020). In the context of cold start, poor results are attributed to the inaccuracy of trained classifiers in capturing uncertainty, a problem that becomes more pronounced with small labeled training sets Nguyen et al. (2015); Gal & Ghahramani (2016). Some recent methods, address issues such as class imbalance, rare classes, and redundancy, *e.g.* in SIMILAR Kothawade et al. (2021). A different category of methods, utilize large batch sizes, aiming to reduce the number of training runs required to update heavy Deep Neural Networks (DNNs). For instance, ClusterMargin Citovsky et al. (2021) addresses the presence of redundant examples within a batch.

The literature suggests only few works for AL in the domain of IIR Barz et al. (2018); Gosselin & Cord (2008); Mehra et al. (2018); Ngo et al. (2016); Rao et al. (2018); Zhang et al. (2002). In this respect, Gosselin & Cord (2008) proposed RETIN, a method that incorporates boundary correction to improve the representation of the database ranking objective in CBIR. In Ngo et al. (2016), the authors introduced an SVM-based Batch Mode Active Learning approach that breaks down the problem into two stages. First, an SVM is trained to filter the images in the database. Then, a ranking function is computed to select the most informative samples, considering both the scores of the SVM function and the similarity metric between the ideal query and the images in the database. A more recent work by Rao et al. (2018) addresses the challenges related to the insufficiency of the training set and limited feedback information in each relevance feedback iteration. They begin with an initial SVM classifier for image retrieval and propose a feature subspace partition based on a pseudo-labeling strategy

Zhang et al. (2002) proposed a method based on multiple instance learning and Fisher information, where they consider the most ambiguous picture as the most valuable one and utilize pseudo-labeling. In contrast, Mehra et al. (2018) adopt a semi-supervised approach for the retrieval model, using the unlabeled data in the pool, for training their retrieval classifier. As for AL, they employ an uncertainty sampling strategy that selects the label of the point nearest to the decision boundary of the classifier, based on an adaptive thresholding heuristic. However, as we argue, uncertainty measures can often be unreliable during cold start, which may limit the effectiveness of this approach. Furthermore, to enhance their results, they incorporate semantic information extracted from WordNet, requiring additional textual input from the user. In comparison, Barz et al. (2018) proposes an AL method called ITAL that aims to maximize the mutual information (MI) between the expected user feedback and the relevance model. They utilize a non-linear Gaussian process as the classifier for retrieval.

Kapoor et al. (2007) introduced an AL technique employing Gaussian processes for object categorization. In each cycle, the method selects a single point-specifically, an unlabeled point characterized by the highest uncertainty in classification. This uncertainty is assessed by taking into account both the minimum posterior mean (closest to the boundary) and the maximum posterior variance. Zhao et al. (2021) introduced an efficient Bayesian active learning method for Gaussian Process classification. In this procedure, one sample is chosen in each cycle. In our method, however, we select a batch of samples by minimizing the overall uncertainty. Additionally, our approach does not rely on knowledge about the distribution of the negative set, which can be highly multimodal due to the presence of various class types.

Several studies have employed a hybrid approach that combines uncertainty and diversity measures Ash et al. (2020); Yang et al. (2015); Karzand & Nowak (2020); Agarwal et al. (2020); Cardoso et al. (2017); Wang et al. (2016). For instance, the BADGE model Ash et al. (2020) is an active learning strategy that uses KMeans++ to create diverse batches, integrating model uncertainty and diversity without requiring hand-tuned hyperparameters—similar to our approach. The CDAL method Agarwal et al. (2020) incorporates spatial context into active detection, selecting diverse samples based on distances to the labeled set and applying this diversity measure within a core-set framework. The CEAL method Wang et al. (2016) proposes Cost-Effective Active Learning by selecting uncertain samples using three common methods: least confidence, margin sampling, and entropy. USDM Yang et al. (2015) introduces a multi-class active learning strategy that explicitly optimizes for uncertainty sampling and diversity maximization, with diversity used as a regularizer. This approach evaluates uncertainty through a generated graph, which limits scalability. Despite their hybrid

nature, these methods are often designed for specific tasks and conditions—such as multi-class, balanced, and closed set scenarios Agarwal et al. (2020); Yang et al. (2015); Ash et al. (2020)—and may not be suited for cold start situations Wang et al. (2016); Ash et al. (2020) or scalable applications Yang et al. (2015) such as those existing in IIR.

Other methods such as MaxMin-based operators focus on the classifier parameters Tong & Koller (2001); Karzand & Nowak (2020). However, these methods are applied to a *single* sample budget, which is associated with increased computation time and user burden due to frequent interactions. Our work is closely related to the MaxiMin algorithm Karzand & Nowak (2020). Nonetheless, we extend and generalize this idea by introducing a flexible framework that can be adapted to different classifiers and accommodate a larger batch size. This is achieved through novel acquisition functions within the proposed greedy method. Ranked Batch-mode Active Learning (RBMAL) Cardoso et al. (2017) takes a different hybrid approach constructing a batches by adding samples with high uncertainty and low maximum similarity to any other already selected sample. We further compare our method to two hybrid methods of MaxiMin Karzand & Nowak (2020) and Ranked batch-mode AL (RBMAL) Cardoso et al. (2017).

## 3 Algorithm Overview and Motivation

This section presents the motivation and key features of our Greedy Active Learning (GAL) algorithm. In the context of a cold start scenario, where the labeled dataset is exceptionally limited, the active learning procedure becomes notably more challenging. The complexity arises from the inability to rely on the classifier to estimate the label or uncertainty of a candidate data point. This scenario is a common challenge in active learning in general Hacohen et al. (2022) and AL-IIR in particular. Additionally, in AL-IIR, there is the open-set classification challenge, involving dealing with unknown classes. The proposed GAL algorithm addresses these challenges through two key aspects: (i) A greedy method that optimally exploits the few labeled samples available and gradually expands the training set within the batch cycle. (ii) Formulating acquisition functions that prioritize data points with the most significant impact on reshaping the decision boundary or the global uncertainty measure. These acquisition functions facilitate improved selection of relevant samples as well as hard-irrelevant (i.e., hard-negative) points that may belong to different unknown categories. Therefore, we depart from the common hypothesis that relies on parameters estimated from a weak classifier (e.g., uncertainty or direct prediction), shifting instead to an approach that focuses on the impact of individual samples on the classifier. This approach is better suited to AL for a binary classifier with few positives and unknown open-set negatives.

In GAL, the samples in the batch are chosen greedily, aiming to maximize an acquisition function that reflects the change in the decision boundary in a MaxMin paradigm. To assess this change, one may require the true labels of the candidate set, which are unavailable in practice. The method therefore calculates a *pseudo-label*, $\hat{l}$, by measuring the change in the decision boundary, for both positive and negative options of each candidate sample. A false label is likely to lead to a larger change in the decision boundary. This is not desirable for the selection, as the importance of the point might be spurious. The true label, though, leads to a smoother and more moderate behavior. This minimal shift, serving as an approximation for the true label, is treated as a pseudo-label. Subsequently, we *maximize* these minimal shifts across the candidates. Figure 2 illustrates with a 2D toy example using an imbalanced dataset with Gaussian distributions in $\mathbb{R}^2$, the rationale behind our pseudo-labeling approach. Positive (relevant) and negative (irrelevant) samples are represented by blue and red colors, respectively. The current training set is depicted in bold, with candidate points shown in a lighter shade. In this scenario, there is one labeled relevant point and 13 labeled irrelevant points. The black dashed line indicates the classifier trained with the whole dataset. Let the dashed green line represent the current boundary (based on the training set). Now, let's select a candidate point (depicted in green). If we designate it as positive and calculate the new boundary, we obtain the blue line; whereas if we designate it as negative, we get the red line. It's important to note that the true label (blue) results in a classifier that is closer to the original dashed green line. Therefore, selecting the label that minimizes the boundary shift approximates the true label. For each candidate point, we determine the pseudo-label and calculate an acquisition function. The optimal point $x^*$ is the one that maximizes this function, resulting in a score which we refer to as an *impact value*. Our algorithm then proceeds to find the next optimal sample. Subsequently, $x^*$ and $\hat{l}^*$ are added to the labeled set. The process repeats to select the next sample until

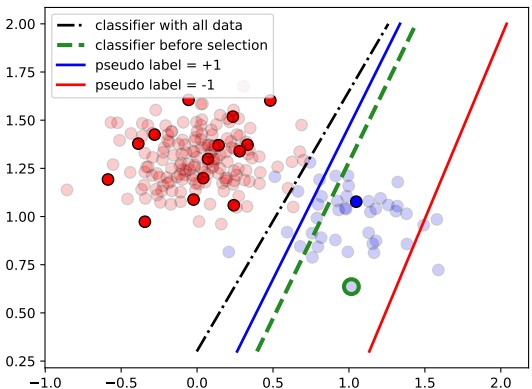

Figure 2: Label proxy demonstration: The points are sampled from two Gaussian distributions, demonstrating the change in the decision boundary for two label options. Red and blue denote negative and positive labels, respectively. Bold and light points represent train and candidate samples, respectively, with their corresponding labels. The green dashed line represents the classifier based solely on the train set (bold circles). The blue and red lines signify the resulting classifier if the selected point (green circle) is labeled as blue or red. The blue classifier exhibits a lower deviation from the dashed green line, consistent with the true label (blue).

a designated budget $B$ is reached. This budget is then allocated for annotation in the next cycle, during which the pseudo-labels are discarded.

We now illustrate the behaviour of various traditional selection strategies, on our toy example in Fig. 3. This toy example demonstrates binary classification in the presence of an imbalanced dataset and a cold-start scenario (consisting of one tagged relevant point and 13 irrelevant points). For each case, we display the current linear classifier (SVM) as a color dashed line and the updated classifier (color solid line) according to different AL selection strategies. For the sake of comparison, we present an upper-bound (in terms of the size of the dataset used for training) of a classifier trained on all the samples with the true labels (dashed black). As observed, random selection achieves a reasonable improvement from the current classifier to the updated version after using the selected points for training. This result is achieved despite ignoring both uncertainty and diversity principles (see also Hacohen et al. (2022)). Kmeans++ is based solely on diversity, selecting points well spread over the dataset. The uncertainty approach (highest Entropy), however, selects points near the current and an inaccurate boundary, caused by the extreme cold-start. Both Kmeans++ and Entropy methods yield an improvement as expected.

However, our greedy method demonstrates the most significant enhancement in narrowing the gap towards the upper-bound classifier. In Fig. 3d, we showcase that our hybrid approach *inherently* incorporates both uncertainty and diversity. The selection sequence ranges from $i_0$ to $i_5$, with $i_0$-$i_2$ and $i_4$ chosen far from the green dashed classifier margin and comply with the diversity principle. On the other hand, two points ($i_3$ and $i_5$) were selected within the classifier margin, tending to comply with the uncertainty principle. Note that, in contrast to Kmeans++ and Random, our approach avoids selecting any irrelevant samples due to an abundance of labeled negatives in the current training set. The combination of our novel acquisition function and greedy approach yields a conditioned diversity, where the diversity depends on the train-set distribution, better coping with the scarcity of labeled samples and the diversity of categories within the dataset.

We further demonstrate this crucial aspect quantitatively in Table 1, where we assess uncertainty, diversity, and accuracy errors. It is evident that Kmeans++ exhibits the highest diversity score, while Entropy demonstrates the highest uncertainty score. GAL, on the other hand, showcases intermediate values and the lowest accuracy error. These metrics confirm that GAL suggests an adaptive strategy that integrates

|  | Uncertainty | Diversity | $\|\theta - \theta_{all}\| \downarrow$ |
|---|---|---|---|
| Random | 0.82 | 0.54 | 1.22 |
| Kmeans++ | 0.86 | **0.78** | 0.98 |
| Entropy | **0.99** | 0.24 | 0.71 |
| SVM-GAL (ours) | 0.96 | 0.52 | **0.29** |

Table 1: Quantitative comparison of diversity versus uncertainty characteristics for various methods with an SVM classifier. Uncertainty represents the mean entropy of the selected points, while diversity denotes the mean pairwise distances among the selected points. The third column indicates the distance between the resulting and the best classifier, where lower values are preferable. Note that GAL achieves a superior balance between uncertainty and diversity factors, effectively addressing both criteria.

both uncertainty and diversity. Throughout the greedy procedure, each subsequent sample is chosen to maximize the impact score, based on the pseudo-labels from the previous samples in the batch. This method avoids choosing samples that have already been selected, as selecting a similar point would not maximize the impact value. Consequently, we achieve the diversity property. Conversely, at certain configurations, the most significant change in the decision boundary is induced by the samples in the classifier margin, specifically those near the boundary with a high level of uncertainty. We also provide an analysis of the cold start performance in Appendix A.

## 4 Algorithm Description

We follow the common strategy in few-shot learning where features are a-priori learned on a large labeled corpus (*e.g.* ImageNet). We then follow the assumption where all the images in the dataset are represented by feature vectors $x_i \in \mathbb{R}^d$, (where $d$ is the feature dimension) either engineered or coming from a pretrained network. In this paper we derive our image features from a pre-trained backbone. Let $\mathcal{X}_u := (x_1, x_2, \ldots, x_m)$ denote the set of *unlabeled* image features (representing the searched dataset), and $\mathcal{X}_l := (x_{m+1}, x_{m+2}, \ldots, x_{m+l})$ the *labeled* set. *Relevant* (positive) and *irrelevant* (negative) samples are labeled by $y_i \in \{+1, -1\}$ respectively, and the label set is denoted by $\mathcal{Y}_l$. The initial labeled set $\mathcal{X}_l$ which defines the query concept, consists of few (usually 1-3) query image features labeled by $+1$. In the course of the iterative process, the user receives an unlabeled batch set $\mathcal{X}_b \subset \mathcal{X}_u$ of size $B := |\mathcal{X}_b|$, and is asked to label the relevant ($y = +1$) and irrelevant ($y = -1$) images. The AL procedure selects the set of $B$ samples, such that when labeled and added to the training set, aims to reach the maximum retrieval performance.

In this work, we suggest a *greedy-based* framework which consists of two phases at each AL cycle. Let $\mathcal{C}_t$ be the classifier at cycle $t$. In the first phase, a candidate subset $\mathcal{X}_c \subseteq \mathcal{X}_u$ of size $K := |\mathcal{X}_c|$ is selected out of the unlabeled pool. This set can be either the whole unlabeled dataset or a subset which is determined by the top-K relevance probabilities. The candidate set $\mathcal{X}_c$ accommodates mostly irrelevant samples due to the natural data imbalance. In the second phase, the algorithm extracts a batch set $\mathcal{X}_b \subset \mathcal{X}_c$ by an AL procedure. A user (oracle) annotates the images selected in $\mathcal{X}_b$ and adds their features and labels into the labeled set $(\mathcal{X}_l, \mathcal{Y}_l)$. Based on the new training set, a classifier $\mathcal{C}_{t+1}$ is trained for the next cycle, as illustrated in Fig. 1.

The selection process aims to choose samples that are most effective when labeled, meaning they maximize the improvement in classifier performance. At each greedy step, an impact value is computed for each unlabeled sample using an acquisition function, which evaluates the sample's contribution to enhancing the classifier. The sample with the highest impact value is then added to $\mathcal{X}_b$ as described in Algorithm 1. We now demonstrate the GAL framework in three settings: linear (SVM) and non-linear (MLP and Gaussian Process) classifiers via the greedy approach.

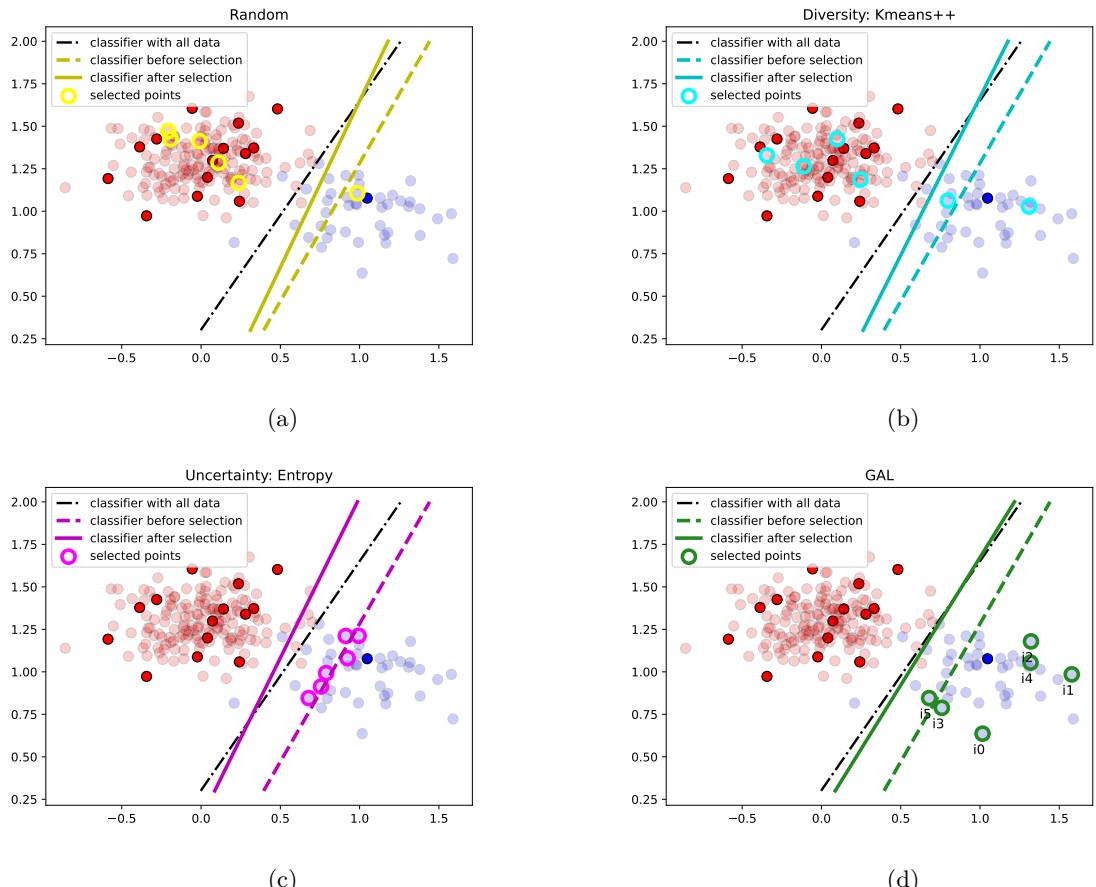

Figure 3: In a 2D Gaussian toy example, we illustrate a binary class scenario characterized by an imbalanced distribution of data, showcasing red samples representing irrelevant data and blue samples representing relevant data. We compare three fundamental selection strategies (a) Random, (b) Pure diversity (Kmeans++), and (c) Pure uncertainty (maximal entropy) to (d), the suggested GAL method. Initially, one relevant and 13 irrelevant samples are labeled. The initial SVM classifier is illustrated by a colored dashed line, followed by the corresponding solid line after updating the classifier with the addition of six samples ($B = 6$). The dashed black line represents an "upper-bound", where the classifier is trained with all the data and their true labels. Notice the most significant improvement observed in the classifier with our GAL method, closing the gap toward the upper-bound and demonstrating a selection pattern that effectively combines diversity and uncertainty. The order of selection in GAL is depicted in (d) by $i_0$ to $i_5$, with corresponding impact scores of 1.75, 1.02, 0.80, 1.06, 0.59, and 0.66. Note that although $i_3$ and $i_5$ are close, they are on opposite sides of the classifier and close to the boundary. This means they have significant uncertainty measures and therefore a substantial impact on the decision boundary.

## 4.1 Sample-wise Impact Value

**Linear Classifier - SVM:** Let us start with a linear classification such as SVM. We define the outcome of a trained *binary* classifier $C$ parameterized by $\theta$, as the measure for the relevance of a sample to a query image. Effective or prominent samples are those that apply the most influence on the classifier's decision boundary. These sample points play a significant role in the active learning process, shaping the classifier's evolution across iterative cycles. However, two primary challenges emerge with this approach: (i) When dealing with a search space that may encompass millions or even more samples, computational efficiency becomes a critical concern. (ii) Due to the scarcity of labels, a shallow classifier such as SVM linear classifier is favored Tong & Chang (2001); Gosselin & Cord (2008); Ngo et al. (2016); Rao et al. (2018). Additionally, SVM has a

---

**Algorithm 1** Greedy Active Learning (GAL) Algorithm

---

**function** GAL($\mathcal{X}_c, \mathcal{X}_l, \mathcal{Y}_l, B$)
    $\mathcal{X}_b \leftarrow \{\}$
    **for** $i \leftarrow 1$ to $B$ **do**
        $x^*, \hat{l}^* \leftarrow \text{NEXT}(\mathcal{X}_c, \mathcal{X}_l, \mathcal{Y}_l)$                        ▷ Find the point that maximizes the impact value $\mathcal{S}$
        $\mathcal{X}_l \leftarrow \mathcal{X}_l \cup \{x^*\}$
        $\mathcal{Y}_l \leftarrow \mathcal{Y}_l \cup \{\hat{l}^*\}$
        $\mathcal{X}_c \leftarrow \mathcal{X}_c \setminus x^*$
        $\mathcal{X}_b \leftarrow \mathcal{X}_b \cup \{x^*\}$
    **end for**
    **return** $\mathcal{X}_b$
**end function**

**function** NEXT($\mathcal{X}_c, \mathcal{X}_l, \mathcal{Y}_l$)
    **for** $i \leftarrow 1$ to $|\mathcal{X}_c|$ **do**
        $x_i \leftarrow \mathcal{X}_c[i]$
        $\mathcal{S}_i, \hat{l}_i \leftarrow \psi(x_i, \mathcal{X}_l, \mathcal{Y}_l)$ by Algorithm 2                    ▷ Acquisition function
    **end for**
    $i^* \leftarrow \arg\max_i \mathcal{S}_i$
    **return** $x_{i^*}, \hat{l}_{i^*}$                  ▷ Return the optimal point and its corresponding pseudo label
**end function**

---

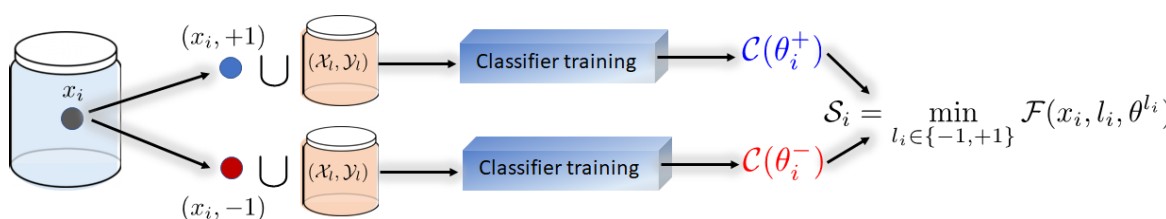

Figure 4: To calculate the score for a point $x_i$ in the candidate set, we train a classifier $\mathcal{C}(\theta_i^+)$ by assuming the sample is positive. Similarly, we train another classifier $\mathcal{C}(\theta_i^-)$ with a negative label. The impact value $\mathcal{S}_i$ is then determined as the minimum value obtained by applying a function $\mathcal{F}$ to both options (4).

strong regularizer to avoid an overfit. Such a classifier also enables relatively rapid training durations. It's important to mention that a single-layer feed-forward neural network (NN) can also be utilized, as it is equivalent to Logistic Regression and is expected to produce outcomes similar to those of SVM. However, the use of a multi-layer perceptron (MLP) for classification carries the risk of overfitting due to the limited size of the training dataset, and potentially resulting in increased computational overhead during the search procedure. We therefore test MLP in the context of AL model (see Sec. 5.2.3). Furthermore, we restrict our examination to samples within the candidate set, denoted as $x \in \mathcal{X}_c$, which is notably smaller than the entire dataset. Regarding the second issue, given the absence of true labels, we employ pseudo labels. The core principle of our proposed algorithm is rooted in the MaxMin paradigm, where we aim to MAXimize the MINimal shift in the decision boundary. This minimal shift serves as an approximation for the true label and is thus treated as a pseudo label.

Let us assume that $x_i$ has a label $l_i$, and $\theta^{l_i}$ represents the parameters of a classifier as if the point $x_i$ is included in the training set with label $l_i$. One possible impact value could be the quantification of the decision boundary's change when $x_i$ is added to the training set. Let $W_0 \in \mathbb{R}^d$ define the initial SVM hyperplane of the AL cycle, and $W \in \mathbb{R}^d$ the hyperplane which was obtained with an additional candidate point $x_i$ with label $l_i$. We then define an acquisition function as

$$\mathcal{F}_{svm} := \|W(x_i, l_i) - W_0\|_2^2. \tag{1}$$

Note that theoretically, there are two unknowns involved in this process. The label, and the most effective point $x^*$ given the label. Ideally, if the labels of the candidate points were known, then

$$x^* = \underset{x_i \in \mathcal{X}_c}{\operatorname{argmax}} \mathcal{F}_{svm}(x_i, l_i, \theta^{l_i}), \tag{2}$$

and $l^*$ is the label of the optimal point. This selection is conditioned on the sample label which is unavailable in practice. We therefore suggest to estimate the label by the minimizer of $\mathcal{F}_{svm}$ such that

$$\hat{l}_i := \underset{l_i \in \{-1,+1\}}{\operatorname{argmin}} \mathcal{F}_{svm}(x_i, l_i, \theta^{l_i}). \tag{3}$$

We refer to $\hat{l}_i$ as a pseudo-label. The acquisition function is therefore defined as

$$\mathcal{S}_i := \mathcal{F}_{svm}(x_i, \hat{l}_i, \theta^{\hat{l}_i}) = \underset{l_i \in \{-1,1\}}{\min} \mathcal{F}_{svm}(x_i, l_i, \theta^{l_i}). \tag{4}$$

The index of the selected point is then given by the largest value among the candidate points,

$$i^* = \underset{i \in 1,2,\dots,|\mathcal{X}_c|}{\operatorname{argmax}} \mathcal{S}_i, \tag{5}$$

where

$$\mathcal{S}_i = \underset{l_i \in \{-1,+1\}}{\min} \mathcal{F}_{svm}(x_i, l_i, \theta^{l_i}). \tag{6}$$

This selection procedure, referred to as NEXT, which utilizes the SVM acquisition function, is detailed in Algorithm 1, the first function in Algorithm 2 and illustrated in Fig. 4.

**Nonlinear Classifier - MLP:** We will now consider a network that comprises of $L$ layers, using a non-linear activation function (ReLU). The classifier is trained using the cross-entropy loss function. As in the linear case, the acquisition function measures the extent of the change in the decision boundary. The AL algorithm remains identical to Algorithm 2, with the only change of replacement of $\mathcal{F}_{svm}$ with $\mathcal{F}_{mlp}$:

$$\mathcal{F}_{mlp} := \|\Psi(x_i, l_i) - \Psi_0\|, \tag{7}$$

where $\Psi$ is a vector of concatenated and flattened network weights. Specifically, $\Psi_0$ defines the initial MLP weights at the current active learning cycle, and $\Psi(x_i, i_i)$ is the weight vector as if the network was trained with $x_i$ and label $l_i$.

---

**Algorithm 2** Acquisition Functions

> **function** $\psi_{svm}(x_i, \mathcal{X}_l, \mathcal{Y}_l)$            ▷ SVM
>      $\theta^+ \leftarrow \text{Classifier}(\mathcal{X}_l \cup x_i, \mathcal{Y}_l \cup +1)$
>      $\theta^- \leftarrow \text{Classifier}(\mathcal{X}_l \cup x_i, \mathcal{Y}_l \cup -1)$
>      $\hat{l}_i \leftarrow \operatorname{argmin}_{l_i \in \{-1,+1\}} \mathcal{F}_{svm}(x_i, l_i, \theta^{l_i})$ by (1) and (6)
>      $\mathcal{S}_i \leftarrow \mathcal{F}_{svm}(x_i, \hat{l}_i, \theta^{\hat{l}_i})$
>      **return** $\mathcal{S}_i, \hat{l}_i$
> **end function**
>
> **function** $\psi_{gp}(x_i, \mathcal{X}_l, \mathcal{Y}_l)$            ▷ Gaussian Process
>      $\mathcal{S}_i \leftarrow \mathcal{F}_{gp}(x_i, \mathcal{X}_l)$ by (13)
>      **return** $\mathcal{S}_i, Null$
> **end function**

---

### 4.1.1 Greedy Approach

The ultimate objective of the AL procedure is to extract a batch consisting of $B$ samples. Ideally, the optimal solution would search for all the permutations of positive and negative labels of the candidate set

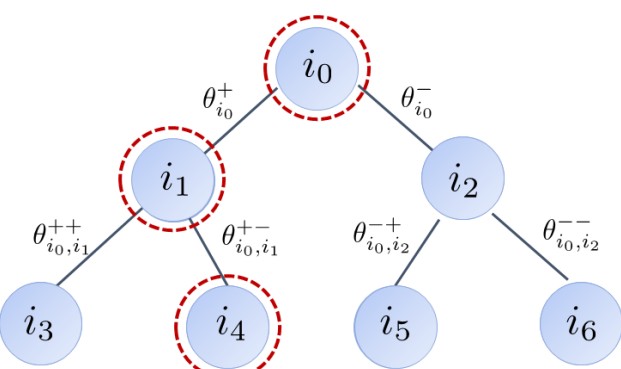

Figure 5: In the SVM scenario, the GAL algorithm employs a binary tree structure. The initial point $x_{i_0}$ is chosen through the NEXT procedure (Algorithms 1). The red circles represent the results obtained from NEXT, which are based on the corresponding pseudo-labels.

such that the impact value would be maximal. This is of course intractable. We therefore use the greedy active learning (GAL) approach which is illustrated in Fig. 5. In GAL, the sample $x_{i_0}$ is initially selected by NEXT (Algorithms 1 and 2). We then insert its pseudo label into the train set, and calculate the next optimal point $x_{i_1}$. In this illustration, $\hat{l}_0 = +1$ associated with the left child of the tree root. At the third iteration $\hat{l}_1 = -1$ and $i_4$ is selected. Samples $i_0, i_1, i_4$ (marked by the red circles in Fig. 5) are then inserted into the budget set $\mathcal{X}_b$. This procedure continues recursively until the budget $B$ is reached, as described in Algorithm 1.

The selection sequence is demonstrated in Fig. 3d. The factors of uncertainty and diversity can drive to different selections. The uncertainty is a by product of the MaxMin operator (5), (6). Points with high uncertainty (close to the boundary) will likely cause the maximum change in the separating hyperplane and therefore will be selected by (1) (see $i3$ and $i5$ in Fig. 3d). As for diversity, selection of nearby samples in the embedding space (which are not close to the boundary) are discouraged due to our approach. Note that whenever a sample point is added to the labeled set, selection of a similar point will result in a low impact value and will be discouraged due to the Max operation, promoting selection of distant points (see the global analysis in Table 1.)

Another theoretical aspect of the algorithm relies on the budget size $B$. The suggested algorithm is highly dependent on the pseudo label $\hat{l}$, where the effectiveness of the AL algorithm increases as the pseudo labels become more reliable. Let $p$ be the probability for a correct pseudo label. The normalized probability, denoted as $P_N$, of obtaining $B$ accurate pseudo labels is given by

$$P_N = \frac{1}{B} \sum_{i=1}^{B} p^i. \tag{8}$$

The normalized probability $P_n$ is plotted in Fig. 6 for different $B$ values and correct pseudo labels probabilities. It naturally suggests that a larger batch size is more sensitive to errors, while a smaller value of $B$ is preferred in each AL cycle. This reasoning will be demonstrated in the experimental results.

### 4.1.2 Complexity for SVM-Based GAL

Lastly, the complexity of training a linear classifier such as SVM is approximately $O(dn^2)$, where $n$ is the number of samples and $d$ is the feature dimension Chapelle (2007). Hence, the complexity of our algorithm at cycle $i$ with $K$ candidates and a budget $B$ is given by

$$\text{Complexity}(i) = \mathcal{O}(BKd(iB)^2). \tag{9}$$

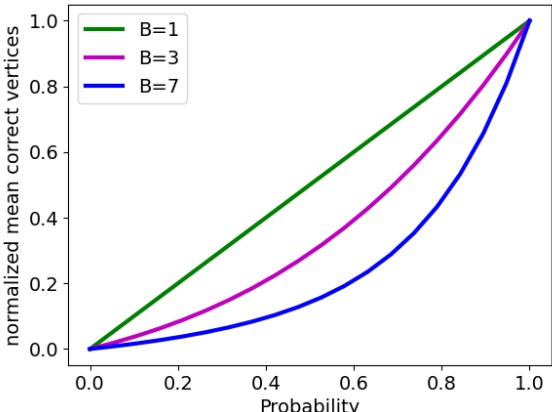

Figure 6: Theoretical results for the normalized probability of obtaining $B$ accurate pseudo-labels vs. the probability of correctly estimating one pseudo-label (see Eq. (8)).

## 4.2 Global Impact Value

**Non-linear Gaussian Process Classifier:** Gaussian Processes (GP) Williams & Rasmussen (2006) are generic supervised learning method designed to solve regression and probabilistic classification problems where the prediction interpolates the observations. Classification or regression by means of a GP, is a non-linear and non-parametric procedure that does not require iterative algorithms for updating. In addition, GP provides an estimate of the uncertainty for every test point, as illustrated in Fig. 7. As can be seen, uncertainty (pink region) is significant as we get further away from the the train (black) points. A Gaussian process can be thought of as a Gaussian distribution over functions $f : \mathcal{X} \to \mathbb{R}$, where in our case $f(x)$ represents the decision boundary. GP is fully specified by a mean function $\mu : \mathcal{X} \to \mathbb{R}$ and a covariance function $\Sigma : \mathcal{X} \times \mathcal{X} \to \mathbb{R}$ (also known as a kernel function). The mean function represents the expected value of the function at any input point, while the covariance function determines the similarity between different input points. The Squared Exponential Kernel is defined as

$$\mathcal{K}(x, x') = \exp\left(-\frac{1}{2\gamma^2}\|x - x'\|^2\right). \tag{10}$$

Let $\mathcal{A} := \mathcal{X}_l$ be the train set of size $L$, and $\mathcal{X}_c$ the candidate set of size $K$. The training kernel matrix is defines as $\Sigma_{11}(\mathcal{A}) \in \mathbb{R}^{L \times L}$ where every entry in the matrix is given by (10) for $x, x' \in \mathcal{A}$. Similarly, the train-test kernel matrix is defined as $\Sigma_{12} \in \mathbb{R}^{L \times K}$, $x \in \mathcal{A}, x' \in \mathcal{X}_c$, and test kernel matrix is given by $\Sigma_{22} \in \mathbb{R}^{K \times K}$, $x, x' \in \mathcal{X}_c$. Then, the mean function is expressed by

$$\mu_{\mathcal{A}} = \Sigma_{12}^T \Sigma_{11}^{-1}(\mathcal{A}) f(\mathbf{x}), \ \mathbf{x} = [x_1, x_2, \dots] \in \mathcal{A},$$

and the covariance matrix is given by

$$\Sigma_{\mathcal{A}} = \Sigma_{22} - \Sigma_{21} \Sigma_{11}^{-1}(\mathcal{A}) \Sigma_{12}. \tag{11}$$

The variance at test point $x'_i$ is given by the diagonal term

$$\sigma_{\mathcal{A}}^2(x'_i) = \Sigma_{\mathcal{A}}[i, i]. \tag{12}$$

Equation (11) reflects the *variance reduction* of the test set due to the train set $\mathcal{A}$. In our setting, $\mu_{\mathcal{A}}(x_i)$ and $\sigma_{\mathcal{A}}^2(x_i)$ denote the decision boundary (red curve in Fig. 7), and uncertainty (pink area in Fig. 7) at point $x_i$ given the train set $\mathcal{A}$. In the AL procedure, our goal is to identify samples that minimize the overall uncertainty. Now, At each AL cycle, if the current train set is denoted by $\mathcal{A}$, we define the acquisition function of a candidate point $x_i$ as the uncertainty area as if $x_i$ was added into the train set,

$$\mathcal{F}_{gp}(x_i) := -\left(\sum_{x \in \mathcal{X}_c} \sigma_{\mathcal{A} \cup x_i}^2(x) + \alpha \max_{x \in \mathcal{X}_c} \sigma_{\mathcal{A} \cup x_i}^2(x)\right). \tag{13}$$

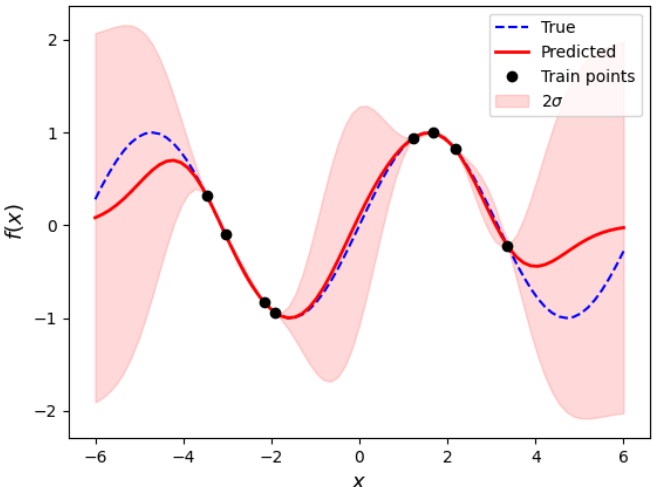

Figure 7: Gaussian Process: The true function is represented by a dashed blue line, while the prediction based on the training points is depicted by the red line. The uncertainty (std) of the prediction is illustrated by the pink area, and the training points are denoted by black circles.

The first term describes the global extent of uncertainty across $\mathcal{X}_c$ in the integral or average sense and is therefore insensitive to abrupt changes in the pointwise variation of $\sigma^2(x)$. On the other hand, the second term represents the $L_\infty$ norm, $\|\sigma^2(x)\|_\infty$ which is designed to manage potential points of discontinuity or large deviations that we aim to minimize. Samples which maximize this function are considered informative[1]. Note that by (11), the uncertainty covariance does not depend on the labels of the training set, avoiding the problem of pseudo labeling. The aqcusition function for the GP is described in the second function in Algorithm 2.

### 4.2.1 Theoretical Analysis

We now investigate the conditions which guarantee a reasonable good approximation to the optimal batch selection. Nemhauser et al. (1978) established a performance lower bound for a greedy algorithm when employed to maximize a set function. Let $B \in \mathbb{N}$ be a budget, $\mathcal{X}$, a finite set and a set function $F(\mathcal{A})$ with $\mathcal{A} \subseteq \mathcal{X}$. For the following maximization problem

$$\mathcal{A}^* = \operatorname*{argmax}_{|\mathcal{A}| \leq B} F(\mathcal{A}),$$

the greedy algorithm returns

$$F(\mathcal{A}_{\text{greedy}}) \geq \left(1 - \frac{1}{e}\right) F(\mathcal{A}^*).$$

under the following conditions:

1. $F(\mathcal{A}) \geq 0$.

2. $F$ is non-negative and monotone, $\mathcal{A} \subset \mathcal{B}$ implies $F(\mathcal{A}) \leq F(\mathcal{B})$.

3. $F$ is submodular if for all subsets $S \subseteq T \subseteq \mathcal{X}$, and all $x \in \mathcal{X} \setminus T$, $F(S \cup x) - F(S) \geq F(T \cup x) - F(T)$.

The submodularity property has the *diminishing returns* behavior: the gain of adding in a particular element $x$ decreases or stays the same each time another element is added to the subset. By (11) and (12), the variance

---

[1]The minus sign is used to change the min to max operator.

at test point $x_i$ is given by

$$\sigma_{\mathcal{A}}^2(x_i) := \Sigma_{22}[i,i] - \left(\Sigma_{21}\Sigma_{11}^{-1}(\mathcal{A})\Sigma_{12}\right)[i,i]. \tag{14}$$

The acquisition function given a train batch $\mathcal{A}$ is then given by

$$F(\mathcal{A}) = -\left(\sum_{x \in \mathcal{X}_c} \sigma_{\mathcal{A}}^2(x) + \alpha \max_{x \in \mathcal{X}_c} \sigma_{\mathcal{A}}^2(x)\right). \tag{15}$$

We now show that the conditions for the $(1-1/e)$-Approximation theorem are satisfied for (15).

The amount of variance reduction for every test point, $\left(\Sigma_{21}\Sigma_{11}^{-1}(\mathcal{A})\Sigma_{12}\right)[i,i]$ is guaranteed to be strictly positive due to the positive-definite nature of the covariance matrix, which is an inherent property of GP modeling, and proved to be increasing monotone and submodular by Das & Kempe (2008). Based on the property that the class of submodular functions is closed under non-negative linear combinations (Fujishige (2005)), (15) is submodular as well. Employing the same considerations implies that (15) exhibits monotonic increasing behavior. Consequently, our acquisition function (15) satisfies the conditions of the $(1-1/e)$-Approximation theorem.

### 4.2.2 Complexity for Gaussian Process-Based GAL

Lastly, the complexity of a matrix of order $n$ inversion is $\mathcal{O}(n^3)$ and two matrix multiplications in (14) are $\mathcal{O}(n^2 K)$ and $\mathcal{O}(K^2 n)$. Hence for each AL cycle $i$ with $K$ candidates and a budget $B$,

$$\text{Complexity}(i) = \mathcal{O}\left(BK\left[(iB)^3 + K^2(iB) + K(iB)^2\right]\right). \tag{16}$$

## 5 Evaluation

Our evaluation includes a comparison with ITAL, a method based on Mutual Information that has been shown to outperform numerous previous approaches. Moreover, we compare our results with several well-established methods in active learning for classification tasks, such as COD, RBMAL, MaxiMin, as well as other strong baselines. We assess the GAL framework by employing three image retrieval techniques, which utilize linear (SVM) and two non-linear (MLP, Gaussian Process) classifiers. The algorithm for SVM and MLP is based on the acquisition functions (1) and (7) respectively. In our evaluation, we compare our approach against various AL algorithms. (i) Random selection, (ii) Cyclic Output Discrepancy (COD) Huang et al. (2021), (iii) MaxiMin Karzand & Nowak (2020), (iv) Ranked batch-mode AL (RBMAL) Cardoso et al. (2017), and in the cases where $B > 1$, (v) Coreset Sener & Savarese (2018); Khakham (2019) and (vi) Kmeans++ Vassilvitskii & Arthur (2006). The COD Huang et al. (2021) method estimates the sample uncertainty by measuring the difference of model outputs between two consecutive active learning cycles,

$$\mathcal{S}_{\text{cod}} := \|\mathcal{C}(x;\theta_t) - \mathcal{C}(x;\theta_{t-1})\| \tag{17}$$

where $\mathcal{C}(x)$ is the classifier prediction, $\theta_t$ and $\theta_{t-1}$ are its parameter set in the current and previous active learning cycles, respectively. MaxiMin Karzand & Nowak (2020) algorithm maximizes the minimum norm of the classifier, *i.e.* prioratizing smoother classifiers among the possible functions

$$\mathcal{S}_{\text{MaxiMin}} := \min_{l \in \{+1,-1\}} \|f(x)^l\|. \tag{18}$$

$\|f(x)^l\|$ denotes the norm of interpolating function when training the classifier with positive and negative labels of $x$. In the linear SVM case, $f(x) = \|W\|_2^2$. RBMAL method Cardoso et al. (2017) combines uncertainty and diversity by

$$\mathcal{S}_{\text{RBMAL}} := \alpha(1 - \phi(x, x_{\text{labeled}})) + (1-\alpha)u(x), \tag{19}$$

where $\phi$ is a similarity measure, $u(x)$ the uncertainty, and $\alpha = |\mathcal{X}_u|/(|\mathcal{X}_u| + |\mathcal{X}_l|)$. The batch set extracted by the above three methods, is obtained by selection of top-$B$ score samples. Kmeans++ Vassilvitskii &

Arthur (2006) and Coreset Khakham (2019); Sener & Savarese (2018) are diversity-based BMAL methods, and therefore applicable for $B > 1$. In Kmeans++, the batch samples are chosen as the closest points to each of the $B$ centroids, and in Coreset, we ensure that the batch samples adequately represent the entire candidate pool based on the $L_2$ norm distance.

In our third image retrieval approach, we incorporate a Gaussian Process (GP) technique, which was proposed in Barz et al. (2018) and referred to as Information-Theoretic AL (ITAL). This method employs a selection strategy that aims to maximize the mutual information between the expected user feedback and the relevance model. To integrate the GP into our framework, we steer the active learning selection process towards data points that minimize the overall uncertainty of the GP classifier, as defined in (13).

## 5.1 Datasets

We evaluate GAL on a wide range of scenarios including 4 datasets, representing image-level and object-level IIR. For instance-level retrieval, we used Paris-6K abbreviated as **Paris**, following the standard protocol as suggested in Radenović et al. (2018). This dataset contains 11 different monuments from Paris, plus 1M distractor images, from which we sampled a small subset, resulting in 9,994 images with 51-289 samples per-class and 8,204 distractors. Next, we built a benchmark based on Places365 Zhou et al. (2017), indicated as **Places**. It contains a larger lake size of 36,500 images with 365 different types of places such as 'restaurants', 'basements', 'swimming pools' etc. Our Places dataset consists of the validation set of Places365. We used 30 classes as queries (randomly sampled) with 100 samples per-class. Lastly, we validated ourselves on object-level retrieval, a previously unexplored task in CBIR-AL. To this end we built a new benchmark from the FSOD dataset Fan et al. (2020), often used for few-shot object detection tasks. At this benchmark, images often include multiple objects (labels), therefore introducing a high challenge for a retrieval model. FSOD dataset is split into base and novel classes. We used the base set, for our benchmark. The base set contains 5, 2350 images with 800 objects categories where each object appears in 22-208 images. As our query pool, we randomly chose 30 object categories appearing in 50-200 images. We refer to this dataset as **FSOD-IR** and we intend to share the protocol publicly for future research.

In all the above experiments, we used a Resnet-50 backbone pre-trained on Imagenet-21K Ridnik et al. (2021). For the first iteration we used the top-K nearest neighbors by the cosine similarity. We used one query for Paris and Places benchmarks, and two queries for FSOD-IR (due to multiplicity of objects in images). We repeated the process for 5 random queries and calculated mAP at each AL cycle. For all these experimetns we used a pretrained ResNet50 features of 2048D.

To ensure a fair comparison between our method and ITAL Barz et al. (2018) and Kapoor et al. (2007), we conducted our evaluation of the GAL framework on the identical dataset of **MIRFLICKR-25K** Huiskes & Lew (2008), which was also employed in ITAL. We followed the same protocol used in ITAL for consistency. This benchmark designed for retrieval consists of 25K images, with query images belonging to multiple categories. We further used the same feature extractor as ITAL (see Barz et al. (2018)). For all datasets we follow the same protocol: sample a query image from a certain class, consider all images belonging to that class (or containing the same object in FSOD-IR) as relevant, while instances from different classes are considered irrelevant.

Our evaluation employs retrieval ranking results, typically measured by mean Average Precision (mAP) Barz et al. (2018); Rao et al. (2018); Mehra et al. (2018); Ngo et al. (2016). In all our experiments, we start with five different initial queries for each class and report mAP as the measure of retrieval performance. According to the standard Interactive Image Retrieval (IIR) process, retrieval is applied to the same corpus at every round, obtaining a new ranked list of results. At each round, the tagged samples are used to update both the retrieval and AL models to be used for the next round. After calculation of mAP at each round we determine the (normalized) area under the curve as the overall score for AL performance.

## 5.2 Experimental Results

We quantified the AL methods by their learning curves, indicating the retrieval performance (measured in mAP) progress along the interactive cycles. The curves are then aggregated by a single measure of the

*Normalized Area under Learning Curve* Barz et al. (2018) between 1,2 to 95 labeled samples. The results for SVM, MLP and GP are averaged over five different randomly selected queries.

As an ablation study, we conducted tests to evaluate the impact of our suggested acquisition functions for AL selection. Additionally, we tested our algorithm under non-greedy settings, indicated as GAL(batch), by selecting the top-$B$ samples that maximize the impact values (1), (6), and (13), given a budget $B$. The non-greedy approach may encounter issues with redundant samples, as similar points could have similar scores. In contrast, the greedy algorithm prevents this scenario by ensuring that once a sample is selected, it is added to the training set. This allows for the selection of a new sample that maximizes the acquisition function, taking into account the updated training set.

### 5.2.1 Runtime and Pool of Selection Candidates

One factor affecting runtime is the ability to achieve a high level of accuracy while searching within a small pool of candidates. We found that selecting from a pool of top-K ranked samples, according to the relevance probabilities obtained from the previous round, is beneficial in GAL and often in competitive methods. This subset $\mathcal{X}_c$ is relatively rich in positive samples and hard negatives, thereby reducing the extreme imbalance in the general dataset. For example, our experiment on FSOD showed that, on average, 30% of the candidate set selected from the top-200 ranked samples were positive, compared to 0.5% in the general dataset. In our model, $K$ can be viewed as a hyper-parameter influenced by the topology of the data in the feature space. The value of $K$ can be estimated through unsupervised analysis of the feature space topology, based on distances from various queries. Conducting this analysis by bootstrapping over randomly sampled queries from our datasets reveals a long tail distribution. We found that typical values around a few percentage of the dataset size (up to 10%), present a reasonable cut-off on this long tail distribution and can also be used to set $K$. Alternatively, $K$ can be set using a different labeled case with the same feature representation (see Sec. 5.2.2). Note that our training process can be easily parallelized by assigning each candidate to a separate process using multi-threading or multi-processing. We report the runtime in Sec. 5.2.2 and Appendix C.

### 5.2.2 SVM Classifier

We first present the global performance measure of *Normalized Area Under Learning Curve* for the SVM-based scenario, tested for budget size $B = 1$ and $B = 3$ in tables 2 and 3. It is worth noting that the results obtained when $B = 1$ allow us to assess the impact value independently from the greedy scheme. We indicate the top performing method in bold and the second place by an underline mark. Interestingly, random sampling often yields high performance. This is consistent to other AL studies in classification benchmarks in the literature, under cold-start conditions Hacohen et al. (2022) (as a diversity based strategy). Yet, in 8 out of 9 tests, GAL outperforms other methods and baselines for $B = 1$, where for $B = 3$, GAL is consistently the top performing method. Note that the top performance for all methods is reached for $K = 100$ or $200$ and there is no consistent competitor in the second place, indicating the robustness of GAL approach under different candidate pools.

Another interesting observation shows that considering a larger candidate pool (from 100 to the whole dataset) does not necessarily improve the performance. Often a smaller candidate pool is preferred as observed in all the methods compared in our datasets for $B = 3$ (cf. Table 2 bottom), due to higher concentration of positive and hard negative samples, being better candidates for AL. For the majority of competitive methods, we discovered that a candidate set size of $K = 200$ is optimal and can significantly reduce the computational cost, an important aspect in an interactive system. The results further show that GAL is relatively insensitive to $K$, above a minimal value, and that this value of $K$ generalizes to other datasets and domains.

Next, we present a comparison of the learning curves by retrieval mean Average Precision (mAP) in figs. 8 and 9 for $B = 1$ and $B = 3$ with $K = 200$. These figures show the superior performance of GAL over previous methods and various baselines. The strongest competitor at $B = 3$ is found to be Kmeans++ which is purely based on diversity, performing comparably to GAL in low the extreme cold start (up to 25 in FSOD-IR and up to 40 in Places). This result is consistent with the analysis in Hacohen et al. (2022) showing that diversity based models such as Kmeans++ or Coreset are top performing methods at extreme cold start. Yet, as more

| Candidate size | Paris | | | | Places | | | | FSOD | | | |
|---|---|---|---|---|---|---|---|---|---|---|---|---|
| | 100 | 200 | 1k | all | 100 | 200 | 1k | all | 100 | 200 | 1k | all |
| Random | 0.847 | 0.942 | 0.834 | 0.810 | 0.375 | 0.390 | 0.298 | 0.224 | 0.576 | 0.630 | 0.452 | 0.404 |
| RBMAL | **0.915** | 0.920 | 0.806 | 0.731 | 0.410 | 0.375 | 0.293 | 0.217 | 0.660 | 0.610 | 0.466 | 0.390 |
| COD | 0.909 | 0.924 | 0.881 | 0.716 | 0.399 | 0.391 | 0.359 | 0.221 | 0.630 | 0.639 | 0.606 | 0.410 |
| MaxiMin | 0.883 | 0.885 | 0.892 | - | 0.395 | 0.381 | 0.363 | - | 0.625 | 0.621 | 0.603 | - |
| GAL (ours) | 0.903 | **0.960** | **0.960** | - | **0.428** | **0.426** | **0.418** | - | **0.674** | **0.672** | **0.672** | - |

Table 2: Normalized Area under Learning Curve with $B = 1$ under different candidate settings. These results indicate the influence of our impact value of the selected samples. We indicate the top performing method in bold and the second place by the underline mark. We omit the test results for "all" in several cases due to increased computation cost and saturation. RBMAL: Cardoso et al. (2017), COD: Huang et al. (2021), MaxiMin: Karzand & Nowak (2020).

| Candidate size | Paris | | | | Places | | | | FSOD | | | |
|---|---|---|---|---|---|---|---|---|---|---|---|---|
| | 100 | 200 | 1k | all | 100 | 200 | 1k | all | 100 | 200 | 1k | all |
| Random | 0.922 | 0.905 | 0.812 | 0.807 | 0.402 | 0.388 | 0.283 | 0.217 | 0.637 | 0.633 | 0.473 | 0.404 |
| RBMAL | 0.923 | 0.888 | 0.785 | 0.718 | 0.397 | 0.355 | 0.295 | 0.213 | 0.652 | 0.592 | 0.467 | 0.389 |
| COD | 0.914 | 0.927 | 0.895 | 0.692 | 0.394 | 0.394 | 0.351 | 0.213 | 0.625 | 0.627 | 0.605 | 0.398 |
| Kmeans++ | 0.922 | 0.941 | 0.935 | 0.744 | 0.416 | 0.417 | 0.394 | 0.205 | 0.661 | 0.666 | 0.632 | 0.393 |
| Coreset | 0.915 | 0.943 | 0.914 | 0.767 | 0.405 | 0.407 | 0.357 | 0.230 | 0.664 | 0.666 | 0.599 | 0.418 |
| MaxiMin | 0.906 | 0.926 | 0.916 | 0.906 | 0.409 | 0.402 | 0.368 | - | 0.657 | 0.648 | 0.612 | - |
| GAL (ours) | **0.946** | **0.960** | 0.952 | - | **0.430** | **0.427** | **0.419** | - | **0.681** | **0.686** | 0.675 | - |
| GAL (batch) | 0.943 | 0.957 | **0.955** | - | **0.431** | 0.421 | 0.417 | - | 0.679 | 0.678 | **0.675** | - |

Table 3: Normalized Area Under Learning Curve with $B = 3$, under different candidate settings. We indicate the top performing method in bold and the second place by the underline mark. GAL(batch) shows the result of our approach without the greedy component of our scheme. RBMAL: Cardoso et al. (2017), COD: Huang et al. (2021), Coreset: Khakham (2019), MaxiMin: Karzand & Nowak (2020).

labels are accumulated, Kmeans++ under-performs GAL that leverages also uncertainty. Furthermore, we note a substantial disparity, with 5-10% (absolute points) higher mAP when compared to MaxiMin (dark green) and around 5% better (from *e.g.* 0.75 to 0.80 in FSOD) compared to Kmeans++.

We conducted an additional investigation using a pure uncertainty-based method, in which the selection criterion involved identifying samples that are positioned closest to the decision boundary. This was achieved by selecting points greedily based on maximum entropy, referred to as *Entropy*. The results for budget size $B = 3$ and $K = 200$ are presented in Table 4. It is evident that the results obtained using this Entropy

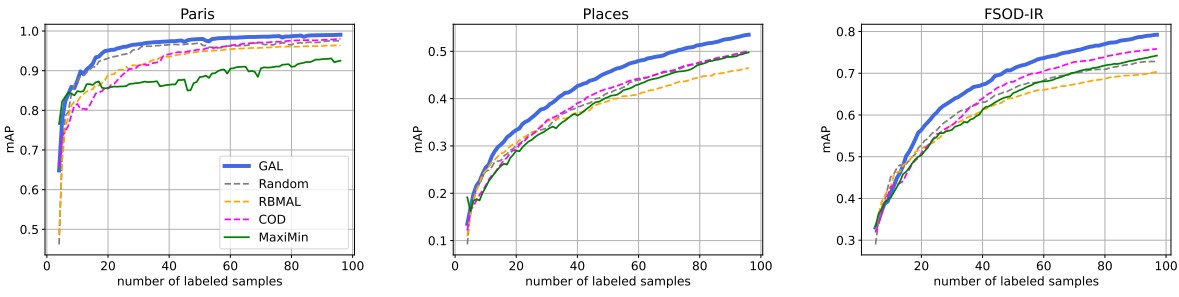

Figure 8: mAP Learning Curves of SVM-based GAL with $B = 1$ and $K = 200$ for different datasets.

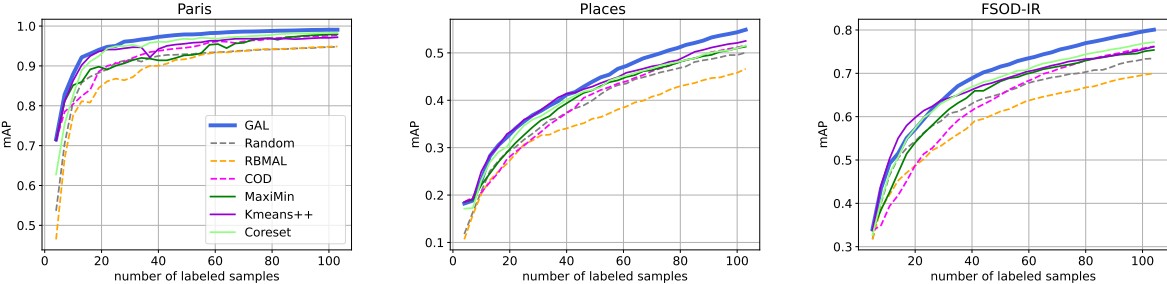

Figure 9: mAP Learning Curves of SVM-based GAL with $B = 3$ and $K = 200$ for different datasets.

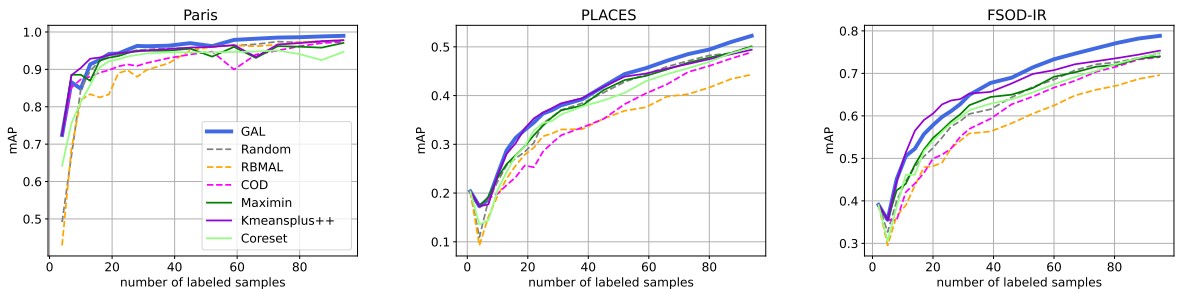

Figure 10: mAP Learning Curves of SVM-based GAL with $B = 3$ followed by $B = 7$ and $K = 200$ for different datasets.

method are considerably inferior to those of GAL across all the datasets. This experiment further strengthens our claim that GAL effectively combines both diversity and uncertainty. Methods that solely rely on one of these aspects tend to exhibit lower performance.

|  | Paris | Places | FSOD |
|---|---|---|---|
| Random | 0.905 | 0.388 | 0.633 |
| RBMAL Cardoso et al. (2017) | 0.888 | 0.355 | 0.592 |
| COD Huang et al. (2021) | 0.927 | 0.394 | 0.627 |
| Kmeans++ | 0.941 | 0.417 | 0.666 |
| Coreset Khakham (2019) | 0.943 | 0.407 | 0.666 |
| MaxiMin Karzand & Nowak (2020) | 0.926 | 0.402 | 0.648 |
| Entropy | 0.903 | 0.329 | 0.586 |
| GAL (ours) | **0.960** | **0.427** | **0.686** |
| GAL (batch) | 0.957 | 0.421 | 0.678 |

Table 4: Normalized Area Under Learning Curve with $B = 3$, $K = 200$. We indicate the top performing method in bold. Entropy shows a selection by the distance to the decision boundary.

As illustrated in Fig. 11a and supported by our earlier analysis presented in Fig. 6, larger budget sizes present more significant challenge, especially during the initial cycles. The challenge is demonstrated in Fig. 11b. During the initial cycles, the pseudo-label accuracy is inadequate, leading to accumulated errors, particularly for larger values of $B$. In response to this challenge, we conducted experiments where we set $B = 3$ for the first 10 cycles, followed by $B = 7$. Nevertheless, our method is superior to other approaches, as shown in Table 5 and Fig. 10. It is noteworthy that overall, although Kmeans++ performed better in the first 10 cycles, our method still showcases superior performance. The greedy approach has a slight impact in the linear SVM case, presumably due to unreliable pseudo-labels, which mostly occur in the initial cycles (see fig. 11b). This strategy is better manifested in the GP process, that is label independent.

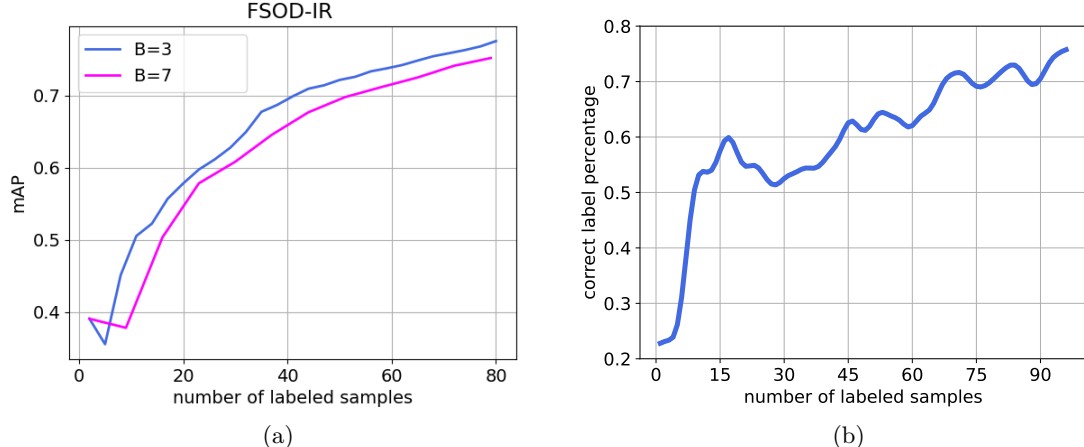

Figure 11: (a) mAP Learning Curves of SVM-based GAL with $B = 3$ and $B = 7$. It is evident that the larger batch size yields inferior results. (b) Pseudo-label accuracy tested on FSOD Benchmark, averaged over all classes and for candidate size of 200 and $B = 1$. Random choice is 50%.

| Candidate size | Paris | | | Places | | | FSOD | | |
|---|---|---|---|---|---|---|---|---|---|
| | 100 | 200 | 500 | 100 | 200 | 500 | 100 | 200 | 500 |
| Random | 0.908 | 0.908 | 0.910 | 0.344 | 0.348 | 0.316 | 0.593 | 0.582 | 0.580 |
| RBMAL Cardoso et al. (2017) | 0.906 | 0.876 | 0.811 | 0.332 | 0.310 | 0.281 | 0.590 | 0.534 | 0.487 |
| COD Huang et al. (2021) | 0.900 | 0.909 | 0.897 | 0.332 | 0.320 | 0.318 | 0.555 | 0.559 | 0.552 |
| Kmeans++ | 0.913 | 0.935 | 0.919 | **0.374** | 0.363 | 0.357 | 0.611 | 0.622 | 0.603 |
| Coreset Khakham (2019) | 0.900 | 0.902 | 0.880 | 0.347 | 0.342 | 0.326 | 0.583 | 0.581 | 0.569 |
| MaxiMin Karzand & Nowak (2020) | 0.910 | 0.925 | 0.919 | 0.355 | 0.353 | 0.323 | 0.589 | 0.591 | 0.563 |
| GAL (ours) | **0.929** | **0.939** | **0.932** | 0.366 | **0.369** | **0.369** | **0.618** | **0.625** | **0.612** |
| GAL (batch) | **0.930** | **0.941** | 0.927 | 0.366 | 0.361 | 0.361 | **0.619** | 0.614 | **0.615** |

Table 5: Normalized Area Under Learning Curve with $B = 3$ at first 10 cycles and then $B = 7$, under different candidate settings. We indicate the top performing method in bold and the second place by the underline mark.

Next, we present a qualitative result displayed in Figure 12. We take two query images belonging to the 'Tin Can' class in the FSOD-IR dataset and showcase the top-16 relevant images retrieved by the GAL and RBMAL methods at the fourth iteration, with a budget of $B = 3$. In the visualization, green and red boxes are used to indicate relevant and irrelevant results, respectively. It's worth noting that the right query image contains not only a 'Tin Can' but also a monitor display. GAL successfully retrieves 15 out of 16 relevant images, with one visually reasonable error. In contrast, the RBMAL method selects a few monitor images, which are exclusively present in the second query image. This example demonstrates a common challenge in CBIR when dealing with images that contain multiple objects. While there may be initial ambiguity in the query, as the active learning cycles progress and the user tags positive examples, our model excels at selecting samples that capture the user intention concept (as shared pattern between the queries) more rapidly.

Finally, despite GAL evaluating a classifier for each selection candidate, the computational cost of our method remains reasonable for several reasons.

1. We demonstrate that a small candidate set of the dataset (obtained from the classifier's top-k), is sufficient as the active learning selection pool. In many cases, this approach even yields improved

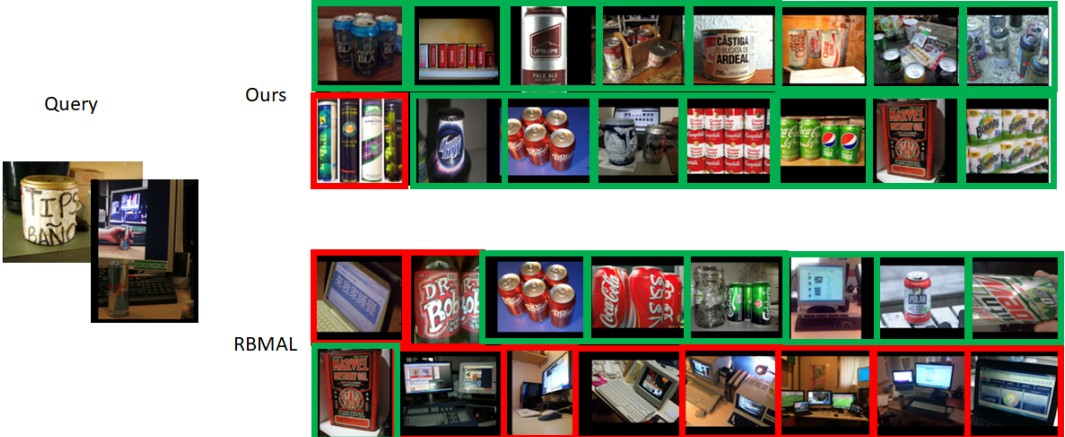

Figure 12: Image retrieval results for *Tin Can* in FSOD-IR dataset with $B = 3$ at iteration 4. Green boxes stand for relevant results while red boxes account for false positives. The second query image has two objects: Can and Display monitor. The RBMAL method mistakenly retrieves images with monitor, where GAL succeeds to find the common pattern in the queries. This example illustrates how the initial ambiguity regarding the object is gradually resolved through the active learning cycles, allowing the algorithm to effectively capture the query concept.

 

    performance, as evidenced in Tables 2 and 3. Consequently, there is no need to run our algorithm on the entire unlabeled set.

2. This allows for quick training and AL cycles, a practical requirement in an interactive system such as IIR.

3. The average runtime for $B = 3$ ranges from approximately 1.2-1.4 seconds per iterations on CPU, for 10 to 30 iterations (without parallelization). In comparison, for MaxiMin, the corresponding times range from 0.5-1 seconds. The remaining faster methods (approximately 0.1 seconds) involve a trade-off in accuracy (see Table 3). We also provide a runtime comparison with ITAL in Appendix C.

### 5.2.3 AL with MLP Classifier

In this section, we present the outcomes of AL when applied to an additional non-linear classifier. It's important to note that the classifier in the context of AL-CBIR comprises two distinct stages: (i) the sample selection strategy (AL) and (ii) retrieval. As discussed in section 4.1, it is crucial to recognize that the utilization of non-linear classifiers in retrieval tasks may lead to immediate overfitting issues, primarily due to the significantly limited size of the training dataset. We therefore extended our work by employing a three-layer MLP (10 neurons at the inner layers) with a ReLU activation function for the AL selection, while continuing to utilize the Gaussian Process (GP) method for retrieval. To make a fair comparison we used the same retrieval method of GP in all compared methods. In this setting as well, the GAL method outperformed competitive algorithms as can be seen in Fig. 13a for the MIRFLICKR dataset with $B = 3$ and $K = 200$.

### 5.2.4 AL with Gaussian Process

We further present the results of GAL utilizing a Gaussian Process (GP) classifier, which are compared to ITAL Barz et al. (2018). For this purpose, we replaced AL module of ITAL with GAL, employing our acquisition function (13). To make a fair comparison, we first ran ITAL with varying candidate pool sizes $K$. Fig. 13b illustrates the results of ITAL for $B = 3$ and $K = 200, 400, 1000$, as well as the entire dataset ($K = 20,000$). We present the results of ITAL for various $K$ settings in Appendix B, showing that the entire unlabeled dataset is needed for ITAL to reach it's best result.

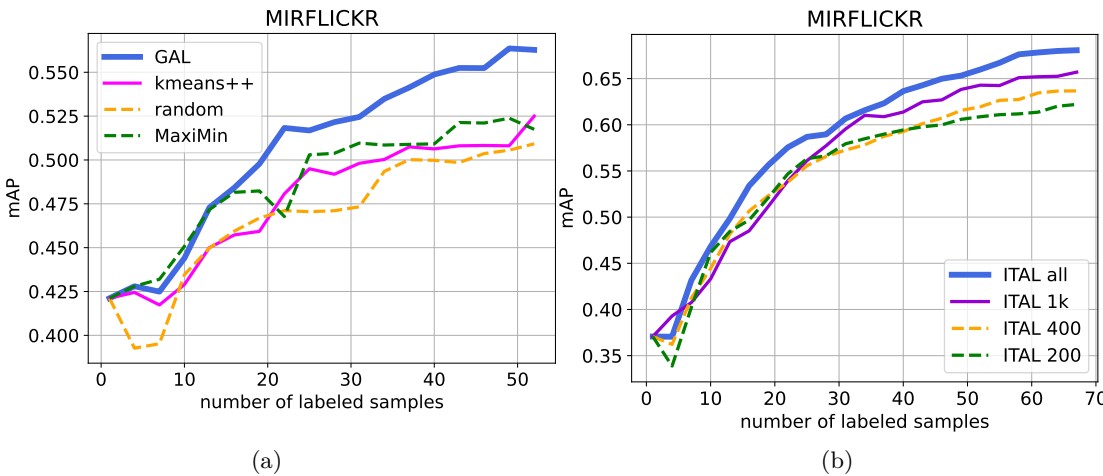

Figure 13: (a) mAP Learning Curves of MLP-based AL selection with $B = 3$ and $K = 200$ applied on MIRFLICKR. (b) mAP Learning Curves of ITAL for $B = 3$ and different candidate set size $K$.

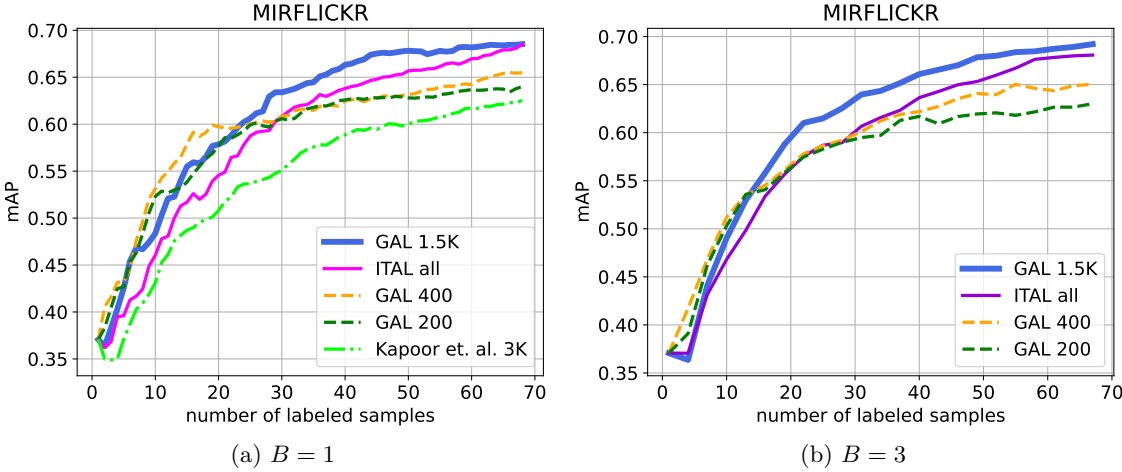

Figure 14: mAP Learning Curves of GP-based GAL with $B = 1$ (a) and $B = 3$ (b) for MIRFLICKR database. ITAL used the whole unlabeled set, while GAL and Kapoor et al. (2007) used different candidate set size K (see Table 6).

Next, we compared GAL and ITAL. Normalized Areas under Curve are summarized in the top panel of Table 6, where GAL outperforms ITAL even when considering only 1,500 points which are 7.5% of the unlabeled dataset as candidates. We further observe the impact of our greedy scheme component boosting the overall performance by nearly 7% (from 0.566 to 0.605) with respect to standard batch selection strategy (denoted by GAL(batch), *i.e.* choosing the top-B scores at each round). Fig. 14 depicts the comparison between these two methods for $B = 1$ and $B = 3$ respectively with candidate pool $K = 200, 400$, and $1, 500$. The figure shows 2-5% mAP improvement with $K = 1, 500$.

Finally, we conducted a comparison between GAL and another uncertainty-based approach proposed by Kapoor et al. (2007) which was designed for $B = 1$. This method aims to identify the sample which is closest to the decision boundary with the highest uncertainty $\sigma$. We adapted this approach to our framework, evaluating its performance across various values of $K$, with the optimal performance observed at $K = 3, 000$. GAL consistently outperformed this method across all tested values of $K$. The summarized results can be found in Table 6 and depicted in the left part of Fig. 14.

| method | $K$ | $B = 1$ | $B = 3$ |
|---|---|---|---|
| ITAL Barz et al. (2018) | 20,000 (all) | 0.586 | 0.585 |
| Kapoor et al. (2007) | 1,500 | 0.517 | |
| Kapoor et al. (2007) | 3,000 | 0.542 | |
| Kapoor et al. (2007) | 20,000 (all) | 0.457 | |
| GAL (ours) | 200 | 0.584 | 0.570 |
| GAL (ours) | 400 | 0.593 | 0.583 |
| GAL (ours) | 1,500 | **0.608** | **0.605** |
| GAL (batch) | 200 | 0.584 | 0.553 |
| GAL (batch) | 400 | 0.593 | 0.573 |
| GAL (batch) | 1,500 | **0.608** | 0.566 |

Table 6: Normalized Area under Learning Curves for MIRFLICKR database. Our GAL outperforms ITAL Barz et al. (2018) and Kapoor et al. (2007). Note that for $B = 1$ there is no greedy process. The impact of our greedy scheme is manifested in $B = 3$.

## 6 Summary and Future Work

In this paper we address the problem of active learning for Interactive Image Retrieval. This task introduces several unique challenges including, a process starting with only few labeled samples in hand and challenging open-set and asymmetric scenario (the negative set includes various unknown categories with different size). In this study, we suggested a new approach that copes with the above challenges by means of two main concepts. First, by considering the impact of each individual sample on the decision boundary as a cue for sample selection in the AL process. To this end, our acquisition functions, may evaluate pseudo-labels or directly optimize a global uncertainty measure. Second, to better cope with the scarcity of labeled samples in a batch mode AL, we embed our approach in a greedy framework where each selected sample in the batch is added to the train set, before selecting the subsequent best promising one. This process is continued until the designated budget is reached, attempting to effectively extend the train set, and provide diversity within each batch. We demonstrate the properties of our method over a toy example, disentangling the two main attributes of AL, namely diversity and uncertainty. We further showed that these attributes are inherently achieved in our approach. Additionally, we provide a theoretical analysis that supports the idea that our greedy scheme offers a reliable approximation (in the context of Gaussian Process). We assessed our approach on various large-scale image retrieval benchmarks, including a new, challenging benchmark featuring small objects. Superior results obtained compared to previous methods, demonstrate the impact of our approach. In addition, we believe that our framework can pave the way for broader applications, particularly, the cold-start problem of AL, in realistic open-set scenarios.

## 7 Acknowledgments

The authors would like to express their gratitude to Nir Sochen and Uri Itai for valuable comments and discussions.

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

# Appendices

## A  Cold Start Analysis

The cold start scenario in active learning refers to the initial phase where the model has a small amount of labeled data to begin with. This lack of labeled data makes it challenging for the model to make accurate predictions or to understand the underlying data distribution. Due tho these challenges, random selection approach is often selected Hacohen et al. (2022). We demonstrate the cold start scenario performance of GAL on the toy example and compare it to the strong baseline of random selection (see discussion in Hacohen et al. (2022) and references therein). Figure A1 shows the mean average precision of our toy example classification (Sec. 3) as the number of labeled points is increased by $B = 2$, up to 100 training samples and 220 test samples. The light and dark blue lines show the performance of the random and GAL algorithms in a cold start, where they start with only 7 labeled points. The light and dark red curves show the results when starting with 50 labeled samples. Clearly, in the cold start case, the classification task is more challenging. Nevertheless, the GAL algorithm outperforms the random selection.

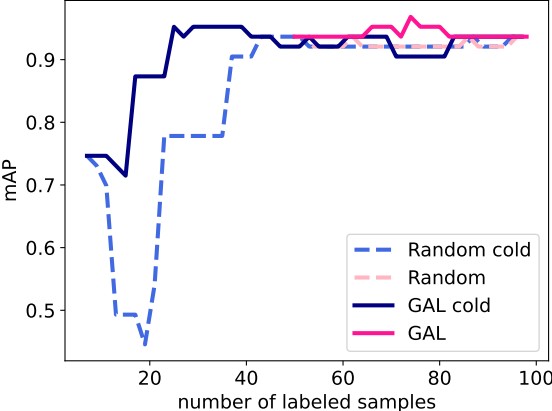

Figure A1: Cold start scenario demonstration. The classification results are shown against the number of labeled samples, where at each iteration, we increased the number by $B = 2$. In the cold start case (light and dark blue), we started with 7 labeled samples, while in the other scenario (light and dark red), we started with 50 labeled samples. In both cases, the GAL algorithm outperforms the random selection.

## B  Analysis of ITAL Performance for Various Candidate Size

Table A1 provides an analysis of ITAL for various candidate set size on MIRFLICKR. It is evident that the entire unlabeled dataset is needed for ITAL to reach the best result.

| $K$ | Normalized AUC |
| --- | --- |
| 200 | 0.547 |
| 400 | 0.552 |
| 1,000 | 0.564 |
| 20,000 | **0.585** |

Table A1: Normalized Areas under Curve of ITAL Barz et al. (2018) for $B = 3$ at variety of candidate set sizes $K$. ITAL requires all the corpus for maximum performance.

## C   Runtime Comparison and Analysis

Here, we show the runtime of GAL-Gaussian Process with ITAL Barz et al. (2018). For a fair comparison, we select the settings for both methods such that they have a similar performance level. Figure A2 illustrates the results, showing that ITAL and GAL have comparable runtime. It is important to note that our training process can easily be parallelized by assigning each candidate to a separate process using multi-threading or multi-processing, thus achieving significant speed-ups.

The experimental data was further fitted to a third-degree polynomial (with respect to $i$ in  (16)) which is in accordance with the complexity equation (16).

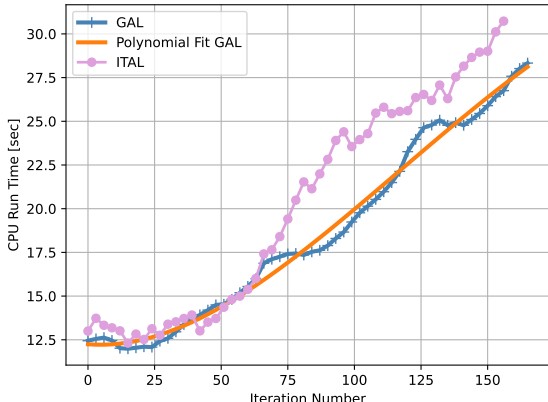

Figure A2: GAL run-time [sec] for GP, $K = 200$, $B = 3$. We further display the run-time of ITAL for comparison, under a setting with the same accuracy level (approximately 0.57), which corresponds to $K = 1000$. GAL shows comparable run-time performance, predominantly due to the fact that GAL needs a significantly lower candidate pool for sample selection. We also demonstrate agreement with the theoretical complexity (16) using a third-degree polynomial fit.

