# OpenReview forum: "Active Learning via Classifier Impact and Greedy Selection for Interactive Image Retrieval"
_TMLR — Accepted by TMLR_

### Review · Reviewer_otno · 2024-05-02

**Summary Of Contributions:**

The paper proposes an innovative batch-mode active learning framework named Greedy Active Learning, designed to address interactive image retrieval. The key innovation of this work is twofold: the introduction of new acquisition functions that quantify the impact value, and the utilization of a greedy scheme to efficiently leverage each sample while coping with a limited number of labeled samples. The proposed framework, which employs different classification methods, outperforms previous approaches and demonstrates strong performance.

**Audience:**

Yes

**Broader Impact Concerns:**

None.

**Claims And Evidence:**

Yes

**Requested Changes:**

See **Weaknesses**.

**Strengths And Weaknesses:**

**Strengths**：
1. The paper effectively conveys ideas, exhibits a well-structured organization.
2. The experimental demonstration section is comprehensive, providing evidence from multiple perspectives to prove the superior performance of the proposed method named Greedy Active Learning.
3. The theoretical analysis section of the paper is meticulously comprehensive, offering conditions that ensure a reasonably accurate approximation to optimal batch selection.

**Weaknesses**
1. In Section 1, the paper acknowledges the existence of hybrid approaches that integrate uncertainty and diversity. It is necessary to introduce their limitations and highlight the advantages of the method proposed in this paper.
2. Some parts of the paper's content are challenging to understand. For example, in Section 1, the statement "On the other hand, diversity is ... the most effective points" requires additional information to enhance readers' comprehension.
3. The paper claims that the proposed active learning method performs well in cold-start scenarios in Section 1. To support this claim, please provide examples of experiments conducted in cold-start scenarios with various sizes of labeled samples.
4. The paper contains some writing errors that need to be corrected. In Section 5, "We asses the ..." should be "We assess the ...". Please check the grammar and writing of the paper once again.
5. The figures in the paper have certain limitations. To prevent distortion, please use vector graphics instead of bitmap images for figures, such as Figure 1, Figure 2, and Figure 7.
6. The bibliography format needs to be corrected to ensure its completeness and consistency.

---

> ### Author Response · Authors · 2024-07-08
> **Detailed response to reviewer concerns**
>
> We sincerely thank the reviewer for her/his valuable remarks and insightful suggestions, which have significantly contributed to improving our paper. We have carefully considered each of their comments and believe that we have now adequately addressed the raised concerns.
> We have modified the paper accordingly and have submitted a revised version. Specifically, we have reorganized the paper and figures, adding appendices to improve the flow of reading. Significant changes and additions are highlighted in blue in the revised version to highlight the changes. Please see our detailed response below.
>
> *1. In Section 1, the paper acknowledges the existence of hybrid approaches that integrate uncertainty and diversity. It is necessary to introduce their limitations and highlight the advantages of the method proposed in this paper.*
>
> We have now added in the newly revised version a dedicated paragraph in Section 2 (Related Work), on page 4, elaborating on this issue
>
> *2.  Some parts of the paper's content are challenging to understand. For example, in Section 1, the statement "On the other hand, diversity is ... the most effective points" requires additional information to enhance readers' comprehension.*
>
> Thank you. We have proofread the article to clarify the content and correct any errors.
>
> *3. The paper claims that the proposed active learning method performs well in cold-start scenarios in Section 1. To support this claim, please provide examples of experiments conducted in cold-start scenarios with various sizes of labeled samples.*
>
> In response, we have added in Appendix A a section that explains and exemplifies the cold start scenario.
>
> *4. The paper contains some writing errors that need to be corrected. In Section 5, "We asses the ..." should be "We assess the ...". Please check the grammar and writing of the paper once again.*
>
> Thank you. We went over the paper again and carefully checked the grammar and corrected the errors.
>
> *5. The figures in the paper have certain limitations. To prevent distortion, please use vector graphics instead of bitmap images for figures, such as Figure 1, Figure 2, and Figure 7.*
>
> Thanks. We have now changed all the figures to PDF format.
>
> *6. The bibliography format needs to be corrected to ensure its completeness and consistency.*
>
> Thank you. We reviewed the references again and made corrections. To shorten the paper length, we used abbreviations for well-known conferences such as CVPR, NeurIPS, and others.

---

### Review · Reviewer_h5uH · 2024-06-15

**Summary Of Contributions:**

The paper introduces Greedy Active Learner (GAL), which is a novel Active Learning (AL) approach for Interactive Image Retrieval (IIR). GAL is robust and efficient when dealing with the challenges of cold-start, open-set, and class-imbalance in a binary classification setup. GAL uses (i) a novel acquisition function for sample selection that measures the impact of each unlabeled sample on the classifier, and (ii) a greedy selection approach that balances diversity and uncertainty within each batch of examples selected for annotation. The paper uses four datasets to evaluate GAL with both linear and non-linear classifiers, and it shows that GAL convincingly outperforms existing approaches.

**Audience:**

Yes

**Claims And Evidence:**

Yes

**Requested Changes:**

The top priority would be to address the two weaknesses above. There are smaller changes suggested below.

p 5: bottom paragraph
- your intuitive argument on why GAL is NOT selecting examples similar to each other is fairly weak. It also seems to be invalidated by Figure 2.d, where there are AT LEAST two pairs of points that appear similar to each other
- afirm --> confirm

p 15: in Fig 13, please add for each graph the horizontal line corresponding to best-known result when training the classifier on all available data; for the downscaled datasets, please also add the same for a classifier trained on the larger, original dataset. W/o these values, it is impossible to judge whether, say,  0.5 on PLACES is a good/mediocre/poor performance (ideally, the learning curve should continue until AL's performance is comparable to supervised training)

p 19: same as above for Fig 15

p 16: except for the first two sentences, the rest of the top paragraph belongs in a DISCUSSION section, rather than in the opening paragraph of the Experimental Results one

**Strengths And Weaknesses:**

The paper has several major strength:
1. Originality - the proposed approach appears to be novel
2. Relevance - active learning is an important area of research with practical applications
3. Future Impact - this paper is likely to have a significant impact on future work on AL
4. Empirical Evidence - the paper provides strong empirical evidence that GAL outperforms existing approaches on the four chosen tasks
5. Overall the paper is well-written and reasonably easy to follow. In particular, Section 3 is extremely helpful in setting up the stage for the intuition behind the proposed. This reviewer wishes that more authors would take this approach in their papers!

The paper also has two main weaknesses:
1. SCALABILITY: the authors should tackle this issue heads on. It is clear that the proposed approach is computationally expensive, so rather than "drop hints" of this limitation (eg, downsizing the Paris-6K dataset, or having un-computed cells in Table 2), the authors should add a section to analyze and discuss this issue
2.  rather than staying with a crisp, simple message of "GAL outperforms its competition," the authors are sometimes trying to make "stretch claims" that - in this reviewer's opinion - hurt, rather than help, this paper. For example, the claim that K=200 is an "optimal size" (bottom of page 17) is later refuted by the evaluation on MIRFLICKR (Table 7); it is also a counter-intuitive claim, especially for open-set, class-imbalanced domains. Even though the observation may be true on these 3 datasets, it is more likely that they all share a peculiar property that makes it so; unfortunately, this claim sounds more like a "diversion" from the bigger issue of scalability (especially in conjunction with  the bullet "1" close to the bottom of page 18). Another example is on page 21, where Fig 17 makes the claim that GAL is faster/comparable to ITAL; if that was truly the case, why not run GAL on all 20K examples?

---

> ### Author Response · Authors · 2024-07-08
> **Detailed response to reviewer h5uH concerns - Part 1**
>
> We sincerely thank the reviewer for her/his valuable remarks and insightful suggestions, which has been instrumental in enhancing the clarity and quality of our paper. We have carefully considered each of the comments and believe that we have now adequately addressed them. Consequently, we have also reorganized the paper and figures, adding appendices to improve the flow of reading. Significant changes and additions are highlighted in blue in the revised version.
>
> 1. *SCALABILITY ...*
>
> Thank you for this recommendation. To address this issue, we have added Section 5.2.1 and Appendix C, which discuss the runtime and highlight GAL's capability to achieve high performance while selecting samples from a small candidate pool of size K. In Appendix C, we show that our approach operates within a reasonable time frame, comparable to the previous ITAL approach, even without parallelization. We believe that multithreading and GPU processing can facilitate operation on large datasets, as our process is inherently parallel (each candidate can be assigned to a separate process), as explained in Section 5.2.2.
>
> Please note that we do not downsize the Paris dataset to reduce runtime. We did it to create a benchmark for active learning in interactive image retrieval task. Furthermore, our Places benchmark is significantly larger, containing 36,500 samples (more details on the dataset sizes have been added to Section 5.1, to clarify this).
>
> 2. *"GAL outperforms its competition”, “stretch claims” ...*
>
> Thank you for pointing this out. We have now moderated our claims to state that our approach has a **runtime** comparable to the previous method of ITAL. We assert that our method can maintain a high performance while having a reasonable runtime, despite its alleged high complexity (analyzed in the paper) because, unlike many others, it does not require processing the entire dataset to achieve a high performance. Our selection strategy effectively identifies valuable samples in the query's close vicinity, selecting valuable positive and hard negatives points. To this end, we present a runtime comparison in Figure A2 in Appendix C (previously Fig. 17) with a configuration where both ITAL and GAL achieve the same level of accuracy. This comparison demonstrates that our computational demand is comparable to ITAL, as we require a significantly smaller pool of candidates, whereas ITAL needs to process the entire dataset to achieve the same performance level. This insight and discussion have now been added to Appendix C.
>
> *For example, the claim that K=200 is an "optimal size" ….*
>
> The value of K depends on the data in the corpus and their feature representation. We have included a discussion in Section 5.2.1 to directly address this issue, exploring ways to select K for a given case. Selecting K can be based on a different labeled dataset with the same feature representation, or through an unsupervised analysis. Achieving top performance with K=200 on Paris, Places, and FSOD datasets, with very different characteristics, implies a generalization capability for setting K, as we mention on page 16 and 17. Note that GAL is also nearly insensitive to $K$ (above a minimal value).
> We discuss these aspects in Section 5.2.1 and on page 16 and 17. Note that the feature representation in MIRFLICKR is quite different, leading to a different K value. The different features in MIRFLICKR were selected to ensure a fair comparison with ITAL, which used these features for their performance assessment.
>
> 3. *p5: bottom paragraph: your intuitive argument on why GAL is NOT selecting examples similar to each other ...*
>
> Thank you for bringing this to our attention. Note that although $i3$​ and $i5$​ are close to each other, they lie on opposite sides of the classifier and near the boundary. The factors of uncertainty and diversity can drive to different selections.  In this case, both $i3$​ and $i5$​ exhibit high uncertainty measures, significantly influencing the decision boundary and outweighing the diversity factor at this particular situation. To provide a measurable comparison beyond visual assessment, we have introduced in section 3 a global quantitative measure for diversity and uncertainty (Table 1). Our findings in Table 1 show that the GAL model achieves a superior balance between these factors, effectively addressing both criteria. We have clarified this point in Sec. 4.1.1 (also referring to it in the caption of Fig. 2).

---

> ### Author Response · Authors · 2024-07-08
> **Detailed response to reviewer h5uH concerns - Part 2**
>
> 4. *p15: in Fig 13, please add for each graph the horizontal line ...*
>
> Please note that the horizontal line, representing the upper bound, is essentially mAP=1. According to the standard Interactive Image Retrieval (IIR) process, retrieval is applied to the same corpus at every round, to obtain a new ranked list of results. Unlike the toy example used for demonstration and visualization, there is no conventional test set here. Following a methodology similar to previous studies, our evaluation is based on the mean Average Precision (mAP) of retrieval results. Each data point on the plot represents an average precision result across various query samples within each class, aggregated over multiple classes (as detailed in the last paragraph of Section 5.1 on page 15).
> As iterations continue, more samples are pulled from the corpus and tagged. These samples are typically positives (relevant) and hard negatives (irrelevant). If the process is continued, at a certain point, all relevant samples in the corpus are likely to appear at the top of the list, resulting in an AP close to 1 (for example see Fig. 10a, on Paris). However, due to the nature of the IIR task, it is reasonable to assume that the user's search for each query will last several rounds and probably up to a few tens of cycles. For this reason, other studies of AL in IIR also do not indicate such a bound.
> In the updated version, we have included an expanded description of our evaluation protocol to allow clarification.
>
> 5.  *p16: except for the first two sentences, the rest of the top paragraph belongs in a DISCUSSION ...*
>
> We have revised this section by consolidating these two sentences into a dedicated special section on Runtime and Pool of Selection Candidates in Section 5.2.1, as requested.
>
> 6. *afirm->confirm, Fig. 14 correct the y-lable*
>
> Thank you. Corrected

---

### Review · Reviewer_aK8m · 2024-10-03

**Summary Of Contributions:**

This paper proposes an active learning method called Greedy Active Learning (GAL) for Interactive Image Retrieval (IIR). AGL addresses two unique challenges: 1) open-set and 2) asymmetric. It uses acquisition functions to quantify the impact value on the classifier based on boundary shit while introducing a greedy method to deal with the scenario where they are a few labeled samples and many open-set samples. The proposed techniques are theoretically justified and empirically verified.

**Audience:**

Yes

**Broader Impact Concerns:**

No ethical concerns are identified.

**Claims And Evidence:**

No

**Requested Changes:**

The proposed method can be better presented. First, summarizes existing works in all relevant fields, the problems they tackle, improvements, and remaining problems. The relationship between AL in image classification vs AL in IIR, how many techniques can be transferred to IIR, and how they perform. Are there any low-scale, mid-scale, and large-scale real-world datasets that can benchmark different IIR AL methods? Why did the authors propose extended datasets in the experiments? The entire methodology can be presented as one single framework, not separated techniques. Improving the presentation and writing quality of the paper.

**Strengths And Weaknesses:**

Strengths:

1. The proposed idea is intuitive and appears to address the active learning problem in IIR well.

2. The proposed techniques seem to be quite effective.

3.  Both illustrative toy examples and theoretical analysis have been provided to help understand the proposed method.

Weaknesses:
1. The paper is poorly written. It is hard to understand the significance of the proposed method to many other methods, like semisupervised learning, unsupervised learning, transfer learning methods, and the SOTA active learning method in IIR. Can the author explain why active learning is the most effective approach for addressing the problem of learning with a few labeled samples and a large amount of unlabeled data?

2. It is hard to tell whether the proposed method works for mainstream models like ViTs. What are the SOTA models for IIR? Can the method work with these SOTA models? Any pre-trained models can be used to help address the problem.

3. The writing of this paper can be improved. The current presentation is under-polished and thus is not ready for publication.

---

> ### Author Response · Authors · 2024-10-14
> **Title: Detailed response to reviewer aK8m concerns - Part 1**
>
> We appreciate the reviewer for the comments and insightful suggestions. We are uncertain whether the reviewer has had the opportunity to read our revised version, as we have previously made revisions and improvements based on the comments provided by the first two reviewers. We have now uploaded a newly revised version. Below, we provide our detailed response to R-aK8m, addressing the concerns raised and outlining the corresponding modifications made to the paper.
>
> *1. It is hard to understand the significance of the proposed method to many other methods…*
>
> One key characteristic of our scenario is the limited amount of labeled data. In Interactive Image Retrieval (IIR), the goal is to progressively improve the retrieval results through user interaction. This interaction can occur through Active Learning (AL), where the system suggests samples from the search data to the user to label as relevant or irrelevant to the query. The objective is to select the most informative samples for training the retrieval model, that will lead to significant retrieval results. As IIR typically starts with only a small set of examples provided by the user, the initial labeled data is limited. During each interactive round, the user labels a few samples to be added to the retrieval model. This setup therefore creates a cold start scenario for AL, and requires the retrieval system to learn from a small training set.
>
> **In terms of comparison with other methods**, please note that this paper specifically focuses on the Active Learning (AL) method, namely the sample selection strategy.  We evaluate the effectiveness of our AL approach by three different classifiers - one linear (SVM) which is mostly used in previous AL-IIR studies, and two non-linear (MLP and Gaussian-Process). We further justify the choice of a simple classifier in terms of both efficiency and accuracy, particularly in scenarios with limited labeled data.
>
> Regarding other **retrieval models** like semi-supervised approaches, we address this in the Related Work section (Sec. 2, 3rd paragraph), where we cite Mehra et al. [33]. Testing additional retrieval methods, such as other semi-supervised techniques, would be interesting (though we believe that unsupervised will be less effective since they ignore the few labels given). Exploring these research topics could be a valuable direction for future research in IIR.
>
> Following the reviewer’s suggestion, we have revised the introduction to provide a more comprehensive and broader overview of Active Learning, AL for IIR and related topics.
>
> *2. The entire methodology can be presented in one single framework…*
>
> Thank you. We appreciate the reviewer's comment. In the revised manuscript, we have reorganized the description into a single framework. The Greedy concept is now presented within a general framework in Algorithm 1. In Algorithm 2, we provide two examples of acquisition functions—specifically, SVM/MLP and Gaussian Process. We believe this structure effectively addresses the reviewer’s concern by presenting our method as a unified framework.
>
>
> *3. It is hard to tell whether the proposed method works for mainstream models …*
>
> Thank you for your feedback. ViT and pretrained models mainly serve as feature extractors in IIR, which is how we use them as well. For retrieval, we choose a simple SVM or GP for two key reasons: First, due to the scarcity of labeled samples, using higher-capacity models like multi-layer perceptrons (MLPs) poses a significant risk of overfitting (note that we demonstrated MLP in a different context of AL strategy in Sec. 4.1). Second, SVMs and GP classifiers allow for fast searches by leveraging efficient algorithms for inner products. This capability ensures reasonable search times, even in large corpora, which is crucial for an interactive system. We address these concerns at the beginning of Sec. 4.1.
>
> *4. Comparison to SoTA IIR methods …*
>
> For comparison to SoTA in Interactive Image Retrieval (IIR) methods, we conducted comprehensive comparisons with the leading works that use AL for IIR in the literature. Section 5 addresses these tests and comparisons.
> While identifying state-of-the-art models for IIR is challenging due to the lack of widely accepted benchmarks in the field, we compared our method with a previous approach of ITAL, which is based on Mutual Information and shown to outperform several previous methods. For a fair comparison, we implemented our method on the dataset and benchmark created by Barz et al. (ITAL). Additionally, we compared our approach with several prominent methods in AL for classification task, including COD, RBMAL, and MaxiMin, as shown in Section 5.2.2 (see Tables).
> We have added a short paragraph to the beginning of Sec. 5 to clarify and emphasize these issues.

---

> > ### Author Response · Authors · 2024-10-14
> > **Title: Detailed response to reviewer aK8m concerns - Part 2**
> >
> > *5. Use of Pretrained models …*
> >
> > The role of a pretrained model is typically in feature representation. We also employ a pretrained ResNet (trained on ImageNet 21K) for feature extraction (see Sec. 5.1).  Our primary focus however, is on the sample selection strategy (AL). We follow established practices to emphasize and demonstrate the contribution and effectiveness of our AL method in improving the sample selection, which ultimately boosts the retrieval performance.
> >
> > *6. First, summarizes existing works in all relevant fields, the problems they tackle, improvements, and remaining problems. The relationship between AL in image classification vs AL in IIR*
> >
> > In response to this concern, we have modified the Introduction section to address broader aspects and the relationship between AL and image retrieval. We begin by introducing the tasks of image retrieval and interactive image retrieval. This is followed by a discussion on the role of Active Learning (AL) in interactive image retrieval. We then highlight the challenges of working with limited labeled samples in both retrieval and AL (cold start problem). Next, we cover relevant fields, discussing the common use case of AL for image classification and the typical conditions under which they operate. This leads to the existing differences between applying AL to IIR and classification. Next, we review specific literature on AL for the IIR task. We further highlight the topic at the beginning of each paragraph to make this more clear.
> >
> > *7. how many techniques [of AL for classification] can be transferred to IIR, and how they perform?*
> >
> > In our evaluations, we included methods commonly used in AL for classification, such as COD, RBMAL, and MaxiMin. However, as we demonstrate in Sec. 5.2, these methods are less effective due to the different characteristics of IIR  (outlined in our paper). Another example is the dominance of random sampling in AL for classification under cold start [1,2,3]. We highlight this in the Introduction (page 2) and include it in our comparisons.
> >
> > [1] Guy Hacohen, Avihu Dekel, Daphna Weinshall, Active Learning on a Budget: Opposite Strategies Suit High and Low Budgets, ICML 2022
> >
> > [2] Kossar Pourahmadi, Parsa Nooralinejad, and Hamed Pirsiavash. A simple baseline for low budget active learning, arXiv:2110.12033, 2021.
> >
> > [3] Akshay L Chandra, Sai Vikas Desai, Chaitanya Devaguptapu, and Vineeth N Balasubramanian. On initial pools for deep active learning. In NeurIPS 2020 Workshop on Pre-registration in Machine Learning, 2021.
> >
> > *8. Are there any low-scale, mid-scale, and large-scale real-world datasets that can benchmark different IIR AL methods?*
> >
> > All of our benchmarks are conducted using real-world datasets. We utilized four datasets: FSOD, Paris, Places, and MIRFLICKR. We believe that the FSOD dataset, which contains 5,350 images featuring 800 objects, can be considered as a low to mid-scale dataset, yet a challenging dataset as it contains several objects in a single image. The Paris-6K dataset, with nearly 10,000 images, can fall into the mid to large-scale category, with a property of retrieving a specific instance. Meanwhile, MIRFLICKR, with 25,000 images, and Places365, containing 36,500 images, can be considered as large-scale datasets, all in comparison to the benchmark sizes often used in related works.
> >
> > *9. Why did the authors propose extended datasets in the experiment?*
> >
> > Sorry, but we are not clear about this question. We have carefully constructed these benchmarks, according to certain characteristics (instance based, high variability, and object based). We have used the images in each dataset without any extension. We will be glad to address this question after further clarification.

---

### Author Response · Authors · 2024-10-14
**Upload of a revised version following reviewer's concerns**

We would like to thank the reviewers for their valuable time and effort. Following the last review (by R-aK8m), we have uploaded a revised version that we believe addresses all reviewer concerns. To make it easier to follow  the changes, we have color-coded most of the modifications made according to  the suggestions received.

We thank R-aK8m for acknowledging the effectiveness of our work, particularly the toy examples and theoretical analysis that contribute to a deeper understanding of our approach. We appreciate again also the positive remarks by other reviewers, who  have recognized the clarity with which our ideas are conveyed, as well as the well-structured organization and thorough experimental demonstration (R-ontno). We also value the Rh5uH positive feedback, that our approach appears novel and holds promise for significant impact on future work in AL, backed by strong empirical evidence and  the paper being well-written.

---

### Decision · Action_Editor_22qL · 2024-11-25

**Recommendation:** Accept as is

**Comment:**

All reviewers recommended acceptance, citing the novelty of the approach, its impact to interactive image retrieval, and solid empirical performance. The authors were responsive to reviewer concerns in their revisions, and reviewers marked their concerns as overall resolved.

**Audience:**

Yes, it is interesting and relevant to TMLR's audience.

**Claims And Evidence:**

This paper presents an active learning method for interactive image retrieval, specifically addressing the issues of open-set and class-imbalanced learning. The approach uses a novel acquisition function for measuring sample importance and a greedy selection mechanism which balances diversity/uncertainty of the labels. The empirical results are strong across multiple benchmarks, with significant improvement over prior work. The initial reviews identified several concerns, including a runtime analysis, clarifications on the theoretical justifications and experiments, scalability, and overall organization of the paper that were addressed in the revisions. Overall, it is a solid contribution, with all reviewers recommending acceptance.